# Wireless neuromodulation in vitro and in vivo by intrinsic TRPC-mediated magnetomechanical stimulation

Chih-Lun Su[1,2], Chao-Chun Cheng[1,2], Ping-Hsiang Yen[1], Jun-Xuan Huang[1], Yen-Jing Ting[1] & Po-Han Chiang [1✉]

Various magnetic deep brain stimulation (DBS) methods have been developing rapidly in the last decade for minimizing the invasiveness of DBS. However, current magnetic DBS methods, such as magnetothermal and magnetomechanical stimulation, require overexpressing exogeneous ion channels in the central nervous system (CNS). It is unclear whether magnetomechanical stimulation can modulate non-transgenic CNS neurons or not. Here, we reveal that the torque of magnetic nanodiscs with weak and slow alternative magnetic field (50 mT at 10 Hz) could activate neurons through the intrinsic transient receptor potential canonical channels (TRPC), which are mechanosensitive ion channels widely expressed in the brain. The immunostaining with c-fos shows the increasement of neuronal activity by wireless DBS with magnetomechanical approach in vivo. Overall, this research demonstrates a magnetic nanodiscs-based magnetomechanical approach that can be used for wireless neuronal stimulation in vitro and untethered DBS in vivo without implants or genetic manipulation.

---

[1] Institute of Biomedical Engineering, National Yang Ming Chiao Tung University, Hsinchu City, Taiwan, R.O.C. [2] These authors contributed equally: Chih-Lun Su, Chao-Chun Cheng. ✉email: phc@nycu.edu.tw

Conventional electrical deep brain stimulation (DBS) has been used for treating neurological disorders, especially motor disorders like Parkinson's diseases, essential tremor and other diseases[1]. However, the use of electrical stimulation requires invasive chronic implantations with electrodes into the deep brain regions[2]. To minimize the invasiveness of DBS, accumulating approaches, including optical[3], acoustic[4] and electromagnetic[5] neuronal modulation approaches, were developed. Optogenetics were using light to activate the opsins on target cell-types. But the lights can be scattered and absorbed easily by biological tissues. The implantation of optical fiber is necessary to deliver lights into deep tissues. The acoustic approach with ultrasound, like sonogenetics and focus ultrasound stimulation, can modulate the neuronal activity without hardware implants. But the ultrasound waves can be scattered, reflected and distorted by skulls and bones. In addition, mounting of ultrasound probes with an aqueous cranial window are required in the acoustic neuronal stimulation. Among all physical approaches, only magnetic fields can penetrate into the brain without absorption or scattering[6]. Transcranial magnetic stimulation (TMS) is a non-invasive neuronal stimulation approach by using strong magnetic fields (>1 T) to induce electric currents in the brain. The strong magnetic fields used by TMS can cause undesirable side effects like muscle twitch, facial pain and other discomforts[7]. The clinical application of TMS is limited to cortical stimulation which cannot be used for DBS. In the last decade, using weak magnetic fields for wireless magnetic DBS were achieved by using magnetic nanoparticle-based neuronal modulation approahces[8–13].

The heat dissipated from magnetic nanoparticles via hysteretic power loss with application of alternative magnetic fields at radio frequency (100 kHz to 1 MHz) were used in magnetothermogenetics[9]. To manipulate the neuronal activity with magnetothermal stimulations, thermosensitive cation channel, transient receptor potential vanilloid 1 (TRPV1), or thermosensitive anion channel, anoctamin1, were overexpressed in the target neurons[9,10,13]. Wireless DBS with magnetothermogenetics have been demonstrated in freely moving mice in vivo. Magnetic DBS at subthalamic nucleus (STN) with magnetothermogenetics could rescue the abnormal behaviors in mice with Parkinson's disease[13]. Lately, another magnetic approach, magnetomechanical stimulation, was demonstrated in both the peripheral nervous system (PNS) and the central nervous system (CNS). In this approach, the mechanical force from the torque of magnetic nanoparticles or magnetic nanodiscs during weak and slow magnetic fields were used to stimulate the neuronal activity wirelessly. In the PNS, mechanosensitive ion channels, Piezo1/2 and TRPV4, are highly expressed in the sensory neurons[14]. A study showed that the torque of ~250 nm magnetic nanodiscs in a weak and slow varying magnetic field (<25 mT at 5 Hz) could induce $Ca^{2+}$ responses in mechanosensitive neurons in primary dorsal root ganglia (DRG)[11]. In contrast to the PNS, the expression level of Piezo1/2 in the CNS neurons is very low. In magnetomechanogenetics, Piezo1 was overexpressed in the target neurons in the brain. Those Piezo1-expressing neurons could be stimulated by the torque of 500 nm magnetic nanoparticles with 20 mT magnetic field at 0.5 Hz[12]. However, in both magnetothermogenetics and magnetomechanogenetics, the potential side effects of overexpressing exogenous genes still remain unknown. The viral vectors for gene delivery in clinical applications also raised safety concerns[15,16]. Therefore, in this study, we developed a non-genetic approach to eliminate the necessity of gene delivery.

The transient receptor potential canonical (TRPC) is a non-selective cation channel family that is highly expressed across various brain regions. There are 3 subgroups: TRPC1/4/5, TRPC2, and TRPC3/6/7. Among them, mammalian TRPC1, 5, and 6 are mechanosensitive and play a role in stretch-stimulated responses[17–19]. TRPC plays multiple functional roles in neurons at both physiological conditions and pathological conditions[20–23], including neuron developments, learning, memory, and fear related behaviors[22,24]. TRPC is involved in various neurological diseases, such as Parkinson's diseases[25], Huntington disease[26], and ischemic stroke[27]. In comparison with Piezo1/2, the TRPC requires higher mechanical force to be activated[28]. It is unclear whether magnetomechanical stimulation can activate intrinsic TRPC. In this study, we hypothesize that intrinsic TRPC-mediated neuronal responses could be triggered by slightly stronger magnetomechanical stimulation than that used for activating exogenous Piezo1 in previous studies[11,12]. Which can be used for neuronal modulation in vitro and in vivo (Fig. 1a). To transduce mechanical force to the neurons, previously described magnetite ($Fe_3O_4$) nanodiscs (MNDs) were used in this study[11]. In absence of magnetic fields, the vortex states of MNDs had better colloidal stability in the solution. In weak magnetic fields at low frequency, the mechanical force generated by the torque of MNDs can activate the mechanosensitive ion channels on the cell membrane[11].

## Results

### Synthesis of magnetic nanodiscs.
Superparamagnetic magnetite nanodiscs (MNDs) were used as nanotransducers for magnetic neuronal stimulations. The anisotropic magnetite nanodiscs were synthesized by a previously described two steps synthesis protocol[11] (Fig. S1a). In the first step, the non-magnetic hematite (α-$Fe_2O_3$) nanodiscs (HNDs) were synthesized by solvothermal method in an autoclave reactor at 180 °C. Next, the MNDs were prepared by reducing the HNDs while maintaining the same sizes. The size of uniform HNDs and MNDs can be tuned by the concentration of $H_2O$ in the solution of first step reaction (Fig. 1b–e, S1b). The diameters of HNDs were 282.8 ± 10.2 nm and 221.0 ± 4.2 nm with respectively 6% and 8% (v/v) $H_2O$ in the first step reaction. The diameters of MNDs were 280.0 ± 5.9 nm and 212.4 ± 7.0 nm when respectively used 6% and 8% (v/v) $H_2O$ in the first step reaction (Fig. 1f-g). The sizes and geometries of HNDs and MNDs from the same synthesis process were similar (Fig. 1f, g, Table S1–3). The X-ray diffraction spectra (XRD) shows that the HNDs from the first step were fully reduced into MNDs (Fig. 1h). Similarly, the magnetization curves show that the HNDs cannot be magnetized with external magnetic field application (Fig. S1c). Contrary to HNDs, MNDs can be magnetized by external magnetic field. The magnetic moments of MNDs calculated from VSM result is $8.7 \times 10^{-16}$ A$m^2$. Therefore, to examine the function of MNDs in neuronal stimulation, the non-magnetic HNDs were used in negative control groups. From calculation in previous study, larger mechanical force could be generated by larger magnetic nanodiscs[11]. Thus, we used nanodiscs with larger diameters (>250 nm) for magnetomechanical stimulation in non-transgenic wild-type neurons (Fig. 1b, c). To functionalize the nanodiscs, all the MNDs and HNDs for neuronal stimulation were coated with poly(maleic anhydride-alt-1-octadecene) (PMAO)[11]. The negatively-charged nanoparticles could promote the attachment of nanodiscs to the excitable neuron cells[29]. The zeta-potential of the PMAO-coated nanodiscs were −53.5 ± 3.7 mV for MNDs and −54.5 ± 2.3 mV for HNDs (Fig. 1i). The PMAO-coated MNDs and PMAO-coated HNDs were used for magnetomechanical neuronal stimulation and the control groups in this study, respectively.

### Magnetomechanical neuronal stimulation with nanodiscs.
Next, MNDs or HNDs (70 μg/ml) were applied on the primary hippocampal cultured neurons. Scanning Electron Microscope

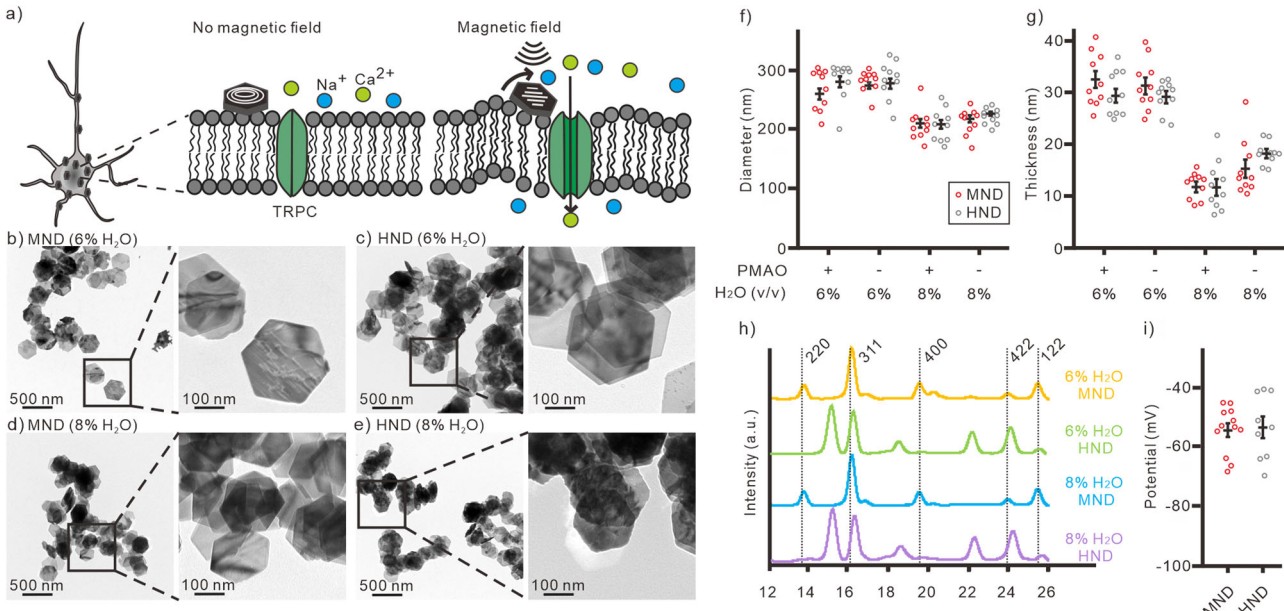

**Fig. 1 Synthesis of MNDs for magnetomechanical stimulation. a** Schematic of magnetomechanical stimulation by using MNDs. TEM images of PMAO-coated MNDs (**b**) and HNDs (**c**) with 6% $H_2O$ in the reaction solution at the first step of reaction. TEM images of PMAO-coated MNDs (**d**) and HNDs (**e**) with 8% $H_2O$ in the reaction solution at the first step of reaction. **f** Diameter of nanodiscs measured from TEM images (All groups, $n = 10$). **g** Thickness of nanodiscs measured from TEM images (All groups, $n = 10$). Statistical analysis of diameter and thickness are shown in Table S1-3. **h** XRD spectra of MNDs and HNDs from different conditions. **i** The zeta-potential of PMAO-coated MNDs ($n = 12$) and HNDs ($n = 9$). $p = 0.917$, Mann-Whitney test. Error bars represent mean ± Standard mean of error (s.e.m.).

(SEM) image shows that both MNDs and HNDs were attached to the membrane of primary hippocampal neurons (Fig. 2a, b). With whole-cell recording, we did not observe any significant difference of the resting membrane potential between neurons with and without negatively-charged MNDs and HNDs (Fig. S2a-d). There were also no significant differences in the other intrinsic properties between groups (Fig. S2e, f). A custom-designed magnetic apparatus for fluorescence microscope was used for magnetic stimulation under upright microscope (Fig. 2c). In the Finite Element Method Magnetics (FEMM) simulation, a homogeneous 50 mT magnetic field was generated within the center of the custom-designed coil with 3 A current application (Fig. 2d, top). Which is very similar to the 50 mT measured from the real coil with 2.8 A current (Fig. 2d, bottom). We didn't observe an obvious increasement of temperature ($\Delta T = 0.20 \pm 0.23$ °C) with the continuous application of 3 A current for 2 min. The neuronal activities were measured by using $Ca^{2+}$ indicator, Fluo-4. We found that when we applied MNDs to the primary cultured neurons, a magnetic stimulation with 50 mT at 10 Hz can induce $Ca^{2+}$ responses in neurons (Fig. 2e, g, h). In contrast, the magnetic field with the same condition cannot induce any $Ca^{2+}$ responses in the control groups with HNDs application (Fig. 2f–h).

To identify the optimal condition for magnetomechanical stimulation with intrinsic mechanosensors, we compared the neuronal responses in the magnetic fields with different frequencies and field intensities. The alternative magnetic fields from 1 to 20 Hz were used to induce $Ca^{2+}$ responses in cultured neurons with MND (70 μg/ml). At different frequencies, the magnetic field intensities were sequentially increased from 10 mT to 50 mT (Fig. 2i–l, S3). We found that the $Ca^{2+}$ responses with 50 mT simulation were significantly larger than other magnetic field intensities from 10 to 40 mT (Fig. 2m). The $Ca^{2+}$ responses at 5 to 20 Hz were significantly larger than responses at 1 Hz. These results indicate that the torque of MNDs was large enough to trigger neuronal activity with 50 mT magnetic stimulation.

When stimulated neurons at 50 mT with different frequencies (Fig. 3a–d), we found that 10 and 20 Hz stimulations induced larger $Ca^{2+}$ responses than 1 and 5 Hz stimulations (Fig. 3e). 10 Hz stimulation could activate more cell population than other conditions (Fig. 3f). In contrast, magnetic stimulation with HNDs application cannot induce any responses in neurons at all conditions (Fig. S4a). These results indicate that 50 mT at 10 Hz was the optimal condition to activate neurons with magnetomechanical stimulation. Moreover, when we applied the magnetic stimulation repeatedly for 4 times, we observed multiple $Ca^{2+}$ responses in cultured neurons at all frequencies (Fig. 3a–d, S4). After repeated magnetic stimulation for 4 times, the cell viability was tested with propidium iodide, a small fluorescent molecule that only can penetrate into the cell membrane of dead cells but not live cells[30]. We didn't observe any cell death after stimulation at different frequencies in either MNDs or HNDs treated neurons. (Fig. 3g, h).

**The magnetomechanical stimulated responses were intrinsic TRPC-dependent.** There are several mechanosensing cation channels expressed in the mammalian cells. including Piezo1/2, TRPC, TRPV4, ASIC3[31,32]. Among them, TRPC and TRPV4 are reported in the CNS[31]. In contrast, Piezo1/2 and ASIC3 are mainly expressed in the PNS[32]. To investigate the mechanism of the MND-mediated responses, pharmacological approach was used to dissect the ion channels that are involved in the magnetic stimulated responses in hippocampal neurons. In TRPC subfamilies, TRPC1, 5 and 6 are reported as mechanosensitive cation channel[17,18]. We found that a specific blocker for TPRC subfamilies, SKF-96365 (50 μM), eliminated the magnetomechanically-induced $Ca^{2+}$ responses (Fig. 4a, e, f, S5a, b). In addition, other non-specific TRPC blockers were also used to exam the contribution of TRPC, including GsMTx4, d-GsMTx4 (5 μM), and 2-Aminoethoxydiphenyl borate (2-APB). GsMTx4 and d-GsMTx4 are antagonists for Piezo1 and Piezo2, respectively. They are also

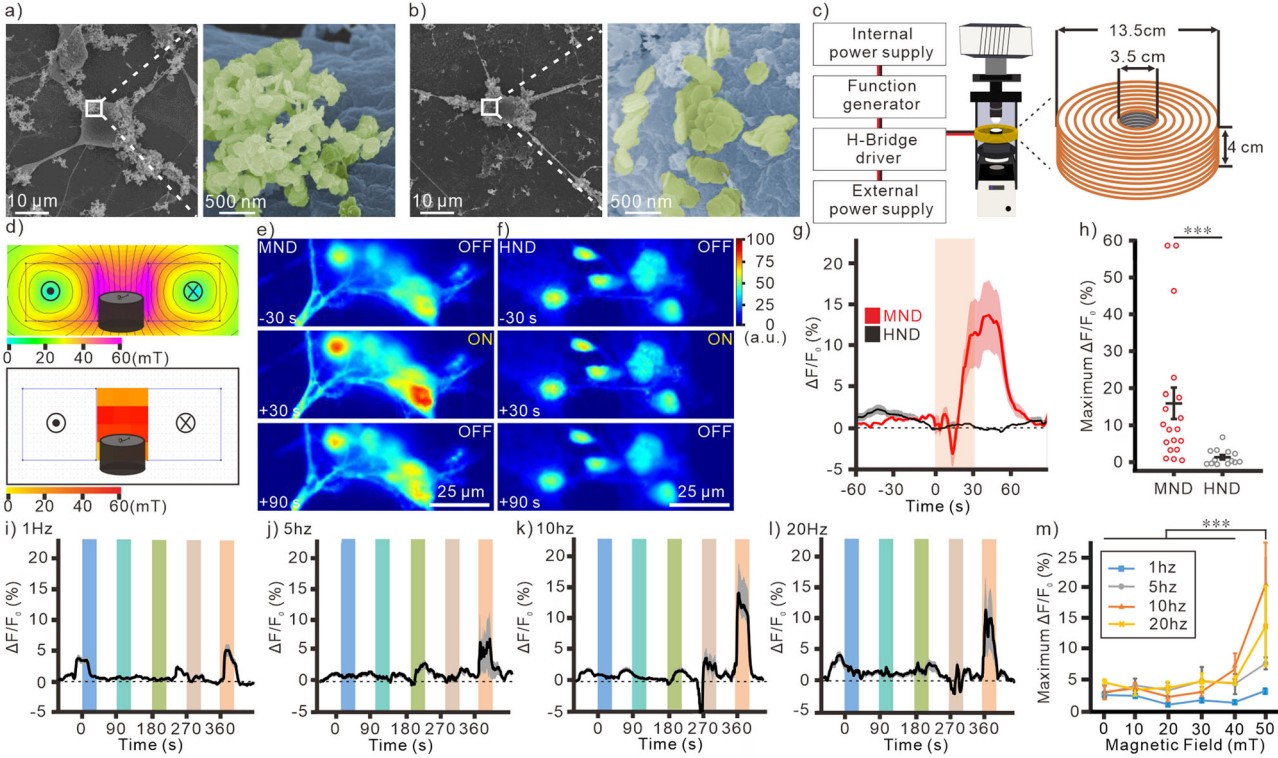

**Fig. 2 MND-mediated neuronal responses by magnetomechanical stimulation.** SEM images of MNDs (**a**) and HNDs (**b**) on the membrane of cultured neurons. **c** Schematic of magnetic apparatus for fluorescence microscope. right, the dimension of the custom-made coil. **d** Heat maps of magnetic fields generated by coil. top, FEMM simulation of the coil with 3 A current. bottom, the measured magnetic field inside of the coil with 3 A DC current. Color maps of fluorescence intensity of neurons with MNDs (**e**) or HNDs (**f**) application. Magnetic field with 50 mT at 10 Hz was applied for 30 s. **g** Average traces of $Ca^{2+}$ responses by magnetic field stimulation. Red line, MND group ($n = 19/3$ (neurons/samples)); Black line, HND group ($n = 13/3$); Light area, s.e.m. **h** Maximum change of $Ca^{2+}$ responses in different groups (MND, $n = 19/3$ (neurons/samples); HND, $n = 13/3$). ***$p < 0.001$, Mann-Whitney test. The averaged traces of magnetic induced neuronal $Ca^{2+}$ responses which were stimulated by different frequencies at 1 Hz (**i**, $n = 46/6$ (neurons/samples)), 5 Hz (**j**, $n = 30/6$), 10 Hz (**k**, $n = 19/6$), or 20 Hz (**l**, $n = 18/6$). In each frequency, magnetic field intensity was sequentially increased from 10 to 50 mT. The gray areas are s.e.m. The color areas indicate the time periods of magnetic stimulation at 10 (blue), 20 (cyan), 30 (green), 40 (pink), and 50 mT (orange). **m** The maximum $\Delta F/F_0$ at different conditions (1 Hz, $n = 46/6$ (neurons/samples)); 5 Hz, $n = 30/6$; 10 Hz, $n = 19/6$; 20 Hz, $n = 18/6$. $F = 9.639$, $p < 0.001$ for field intensity, $F = 6.155$, $p < 0.001$ for frequencies; $F = 1.459$, $p = 0.11$ for interaction of frequencies and intensities. Two-Way ANOVA. ***$p < 0.001$, Tukey post-hoc test. Error bars represent mean ± s.e.m.

antagonists for TRPC1,5 and 6[19]. 2-APB is TRPC antagonist and TRPV1-3 agonist, but insensitive to TRPV4[33]. Similar to SKF-96365, all the TRPC blockers, including GsMTx4 (5 μM), d-GsMTx4 (5 μM), and 2-APB (100 μM), were able to eliminate the magnetomechanically-induced responses in hippocampal neurons (Fig. 4b–f, S5a, b). These results indicate that intrinsic TRPC in hippocampal neurons plays critical roles in magnetomechanical stimulations.

Next, the role of TRPV4, another mechanosensitive ion channel, in magnetomechanical stimulation in hippocampal neurons was investigated by applying TRPV4 specific blocker, HC-067047[11]. We found that HC-067047 (1 μM) couldn't modulate the MND-mediated responses (Fig. 5a, f, g, S5c, d). It indicates that the magnetic stimulated responses were not mediated by TRPV4. The voltage-gated sodium channel (VGSC) blocker, tetrodotoxin (TTX; 100 nM), was used for investigating whether action potentials were involved in the magnetomechanically-induced $Ca^{2+}$ responses. The first stimulation induced $Ca^{2+}$ responses and cell activities are almost eliminated with TTX application (Fig. 5b, f, g). This result indicates that the magnetomechanical stimulated TRPC activation could induce action potentials in neurons. Interestingly, the maximum fluorescence changes and cell activity ratio of multiple stimulation were significantly reduced but not completely abolished by TTX application (Fig. S5c, d). These TTX-

independent $Ca^{2+}$ responses might be contributed by other action potential-independent $Ca^{2+}$ channels. TRPC is known as a non-specific cation ion channel with $Ca^{2+}$ permeability. The TTX-independent $Ca^{2+}$ responses might be contributed by TRPC alone or by other voltage-gated $Ca^{2+}$ channels (VGCC). Among all VGCC, T-type VGCC and $Ca_V1.3$ of L-type VGCC are low-threshold-activated ion channels. To investigate whether VGCC are involved in magnetomechanical stimulated responses, antagonists of L-type and T-type VGCC were applied during multiple magnetomechanical stimulations. Nifedipine and Mibefradil are commonly referred as a specific L-type VGCC antagonist and a specific T-type VGCC antagonist, respectively. However, accumulating evidence shows that they are non-specific antagonists which can block both L-type and T-type VGCC[34]. By using nifedipine (10 μM)[35], there were almost no $Ca^{2+}$ responses with multiple magnetic stimulations (Fig. 5c, f, g, S5c, d). Similarly, with mibefradil (3 μM)[36] application, $Ca^{2+}$ responses by multiple magnetic stimulation were significantly reduced (Fig. 5d, f, g, S5c, d). These results indicate that the magnetomechanically-induced TRPC activation could trigger VGCC. The TTX-independent $Ca^{2+}$ responses might not be contributed by $Ca^{2+}$ influx through TRPC alone. Finally, with $Ca^{2+}$-free extracellular solution, all the magnetomechanically-induced $Ca^{2+}$ responses were eliminated (Fig. 5e–g, S5c, d). These results indicate that these $Ca^{2+}$ responses were contributed

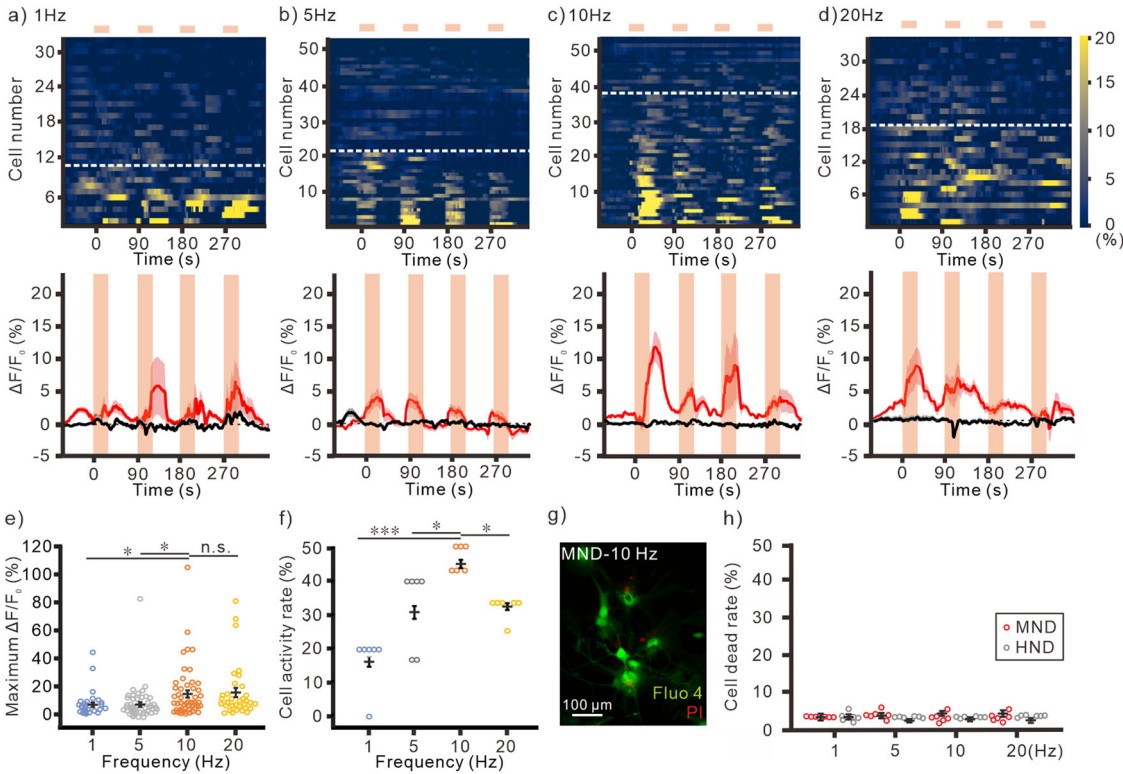

**Fig. 3 Tuning the magnetic field frequency for wireless stimulations.** Multiple stimulations with 50 mT at 1 Hz (**a**, $n = 32/6$ (neurons/samples)), 5 Hz (**b**, $n = 51/6$), 10 Hz (**c**, $n = 52/6$), or 20 Hz (**d**, $n = 34/6$). top, Heatmap of $Ca^{2+}$ responses in individual neurons. The cells below white dash lines were activated during stimulations. bottom, The averaged traces of fluorescence change. The light orange areas indicate the time periods of magnetic stimulation. Red line, MND group. Black line, HND group. Pink and gray area, s.e.m. **e** The maximum change of fluorescence at different frequencies (1 Hz, $n = 32/6$ (neurons/samples); 5 Hz, $n = 51/6$; 10 Hz $n = 52/6$; 20 Hz, $n = 34/6$). $F = 1.050$, $p < 0.001$, Kruskal-Wallis test. **f** The cell activity rate of each culture sample at different frequencies (All groups, $n = 6$). $F = 8.208$, $p < 0.001$, Kruskal-Wallis test. *$p < 0.05$, ***$p < 0.001$, Dunn post-hoc test. **g** PI treatments in neurons with MNDs after magnetomechanical stimulation at 10 Hz. Green, Fluo-4; Red, PI. **h** Quantify of live-dead assay with stimulations at different frequencies (All groups, $n = 6$). $F = 0.003$, $p = 0.995$ for frequency; $F = 0.003$, $p = 0.862$ for type of nanodiscs; $F = 0.022$, $p = 0.995$ for interaction between frequencies and nanodiscs, Two-way ANOVA. Error bars represent mean ± s.e.m.

by $Ca^{2+}$ influx from the external solution (Fig. 5h). Overall, from the pharmacological study, we found that the torque of MND generated by an alternative magnetic field can induce $Ca^{2+}$ influx from external solution. Which were mainly caused by VGCCs and action potentials in cultured neurons. The SKF-96365, GsMTx4, d-GsMTx4 experiments show that this magnetomechanical stimulated responses were mainly mediated by intrinsic TRPC (Fig. 4a–f, S5a, b).

**Wireless DBS by magnetomechanical stimulation in vivo.**
Subthalamic nucleus (STN) is the clinical target of conventional DBS with electrical stimulation for treating patients with Parkinson's diseases[37]. By using immunohistochemistry, we observed that all the mechanosensitive TRPC, including TRPC1, 5, and 6, were expressed in the STN of mice (Fig. 6a–c). These results are similar to a previous report of TRPC expression in STN of rats[38]. To investigate the magnetomechanical neuronal modulation for DBS in STN in vivo, nanodiscs (2 μl of 1 mg/ml) were unilaterally injected into STN of mice (Fig. 7a). The coordinate of injection was confirmed by using fluorescence-labeled MNDs (Fig. S6a, b). After injection, the fluorescence-labeled MNDs were not degraded or metabolized for at least 5 days (Fig. S6c, d). For wireless neuronal stimulation, PMAO-coated MNDs and HNDs without fluorescence were injected into the same coordinate at STN. At 5 to 7 days after injection, the nanodiscs injected mice were placed into a large custom-designed magnetic apparatus with

20 cm inner-diameter and 25 cm height (Fig. 7b). This magnetic apparatus consisted of 4 round coils (20 cm inner-diameter, 28 outer-diameter and 5 cm height) controlled by 4 sets of drivers and power supplies. The FEMM simulation demonstrated that the magnetic field in the center of the coil with 10 A current is 50 mT (Fig. 7c). The magnetic field measured from the center of the custom-made coil is 50 mT with 10 A current application (Fig. 7d). The temperature of the coil increased less than 1 °C when applied 10 A current for 2 min ($\Delta T = 0.70 \pm 0.15$ °C at the wall; $\Delta T = 0.83 \pm 0.14$ °C at center). The awake mice were stimulated by the magnetic field with 50 mT at 10 Hz with 30 s on-30 s off cycle for 10 min (Fig. 7a, b). We found that the immediate early gene, c-fos, expressions in MNDs injected STN were significantly larger than contralateral STN (Fig. 7e–g). In contrast, there was no difference between the c-fos expressions of ipsilateral and contralateral STN of HNDs injected mice (Fig. 7h, S7a–b). The ipsilateral/contralateral ratios of c-fos expressions in STN of MNDs injected mice were also significantly more than HNDs injected groups (Fig. 7i). Entopeduncular nucleus (EP) is one of the downstream of STN glutamatergic projecting neurons in mice. And it is homologous to internal Globus Pallidus (GPi) in humans. Similar to STN, c-fos expressions in the ipsilateral EP of MNDs injected mice were significantly more than contralateral EP (Fig. 7j, S7c, d). But in HNDs injected mice, there was no increase of c-fos expression in EP (Fig. 7k, S7e, f). The ipsilateral/contralateral ratios of c-fos expressions in EP of MND injected mice were also significantly more than HNDs injected groups

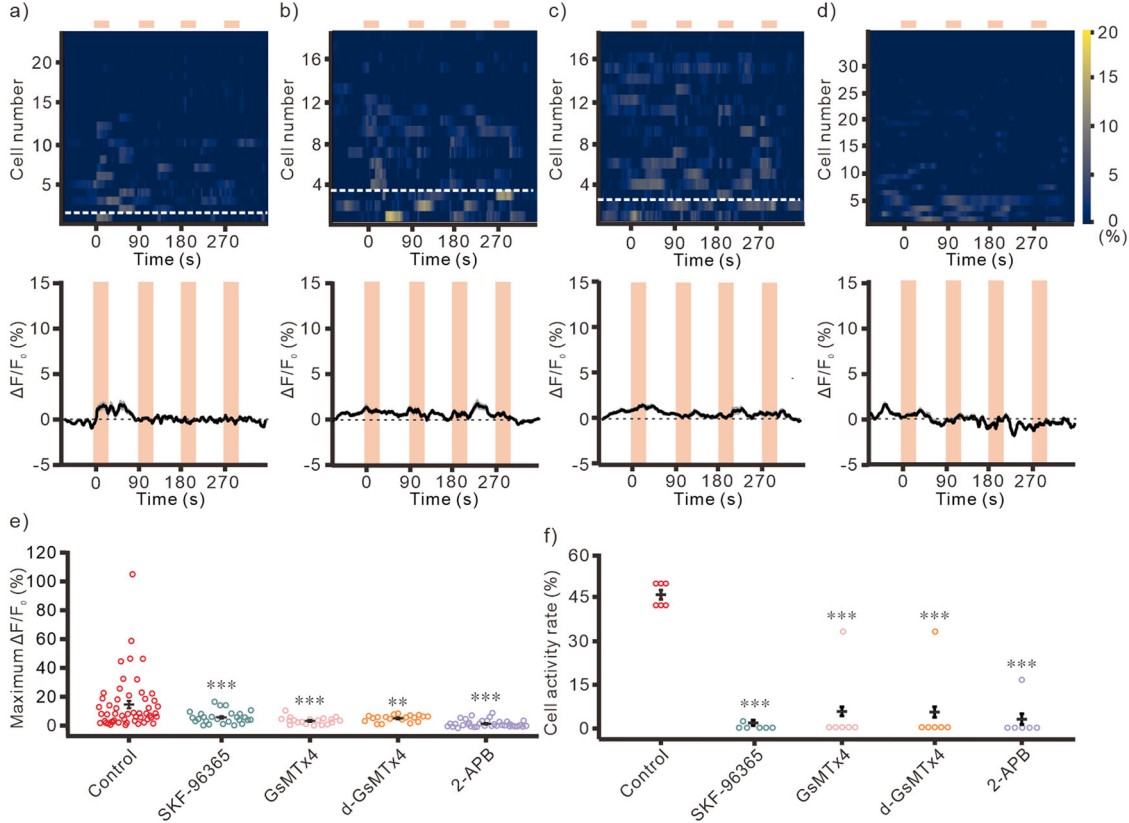

**Fig. 4 Reducing the magnetomechanical stimulated responses by different TPRC antagonists.** The $Ca^{2+}$ responses by multiple magnetomechanical stimulation with bath application of SKF-96365 at 50 μM (**a**, $n = 26/6$ (neurons/samples)), GsMTx4 at 5 μM (**b**, $n = 19/6$), d-GsMTx4 at 5 μM (**c**, $n = 19/6$), and 2-APB at 100 μM (**d**, $n = 36/6$). The magnetic stimulations are 50 mT at 10 Hz for 30 s. Inter-stimulation intervals are 60 s. top, heatmap of individual cells responses. The cells below white dash lines were activated during stimulations. bottom, averaged fluorescence changes. The light orange areas indicate the time periods of magnetic stimulation. The gray areas are s.e.m. **e** Maximum fluorescence changes of MNDs treated neurons with different TRPC blockers at the first magnetic stimulation (Control, $n = 52/6$ (neurons/samples); SKF-96365, $n = 26/6$; GsMTx4, $n = 19/6$; d-GsMTx4, $n = 19/6$; 2-APB, $n = 36/6$). $F = 146.799$, $p < 0.001$, Kruskal-Wallis test. **f** The cell activity rate of MNDs treated neurons with different TRPC blockers at the first magnetic stimulation (All groups, $n = 6$). $F = 29.017$, $p < 0.001$, Kruskal-Wallis test. $**p < 0.01$, $***p < 0.001$, compared to the control group; Dunn post-hoc test. Error bars represent mean ± s.e.m.

(Fig. 7l). In previous studies, when unilateral stimulating STN of healthy wild-type mice without Parkinson's diseases, the rotation behavioral results are controversial. Here, we didn't observe obvious rotation behaviors in awake mice with wireless magnetic stimulations in these conditions (Fig. S8). The results of c-fos staining show that by using the magnetomechanical approach with MNDs, we were able to wirelessly modulate the neuronal circuit in the deep brain region in vivo.

## Discussion

In conclusion, we found that when applied weak and slow alternative magnetic fields (50 mT at 10 Hz), the mechanical force generated by magnetic nanodiscs induced the activity of wild-type neurons without overexpressing exogenous genes. We reveal that these magnetomechanical responses were mainly mediated by the intrinsic mechanosensitive cation channel, TRPC. Finally, the activities of deep brain regions were increased by MND-mediated magnetomechanical stimulation in awake mice in vivo. Recent studies of magnetomechanical stimulation in Piezo1-expressing DRG or overexpression Piezo1 in CNS only require <23 mT at 1~5 Hz. In line with previous reports, we did not observe obvious responses with magnetic fields at <40 mT in neurons without Piezo1/2 or TRPV4 overexpression (Fig. 2i–m). The magnetic field intensity for inducing TRPC activity in our study are larger

(50 mT) than previous research. Which is in accordance with the previous report that TRPC requires stronger mechanical force than Piezo1[28]. However, we cannot rule out the possibility that TRPC might be activated by mechanical stimulation indirectly[39] when we apply magnetomechanical stimulation with MNDs.

The MNDs and HNDs in this study were functionalized with PMAO and were carrying negative surface charges (Fig. 1i). Previous study shows that the negatively-charged nanoparticles can facilitate the attachment of the particle on the excitable neuronal membrane but not on the non-excitable glial membrane[11,29]. In contrast, the positively-charged or neutral nanoparticle cannot attach to the neuronal membranes[29]. Although the negatively-charged nanoparticles at the surface of the neuronal membrane might increase the excitability[29], we didn't observe significant increasement of resting membrane potential and neuronal activity both in vitro and in vivo. First, there are no $Ca^{2+}$ responses in cultured hippocampal neurons with negatively-charged HNDs (Fig. 2e–h). With whole-cell recording, the resting membrane potentials of MNDs-treated and HNDs-treated neurons were not significantly higher than nanodiscs-free control groups (Fig. S2a–c). Finally, in c-fos immunostaining, the neuronal activities in vivo in the STN and its downstream EP of unilateral HNDs injected mice had no differences between ipsilateral and contralateral hemisphere (Fig. 7h, k). Nevertheless, further modification of the surface of

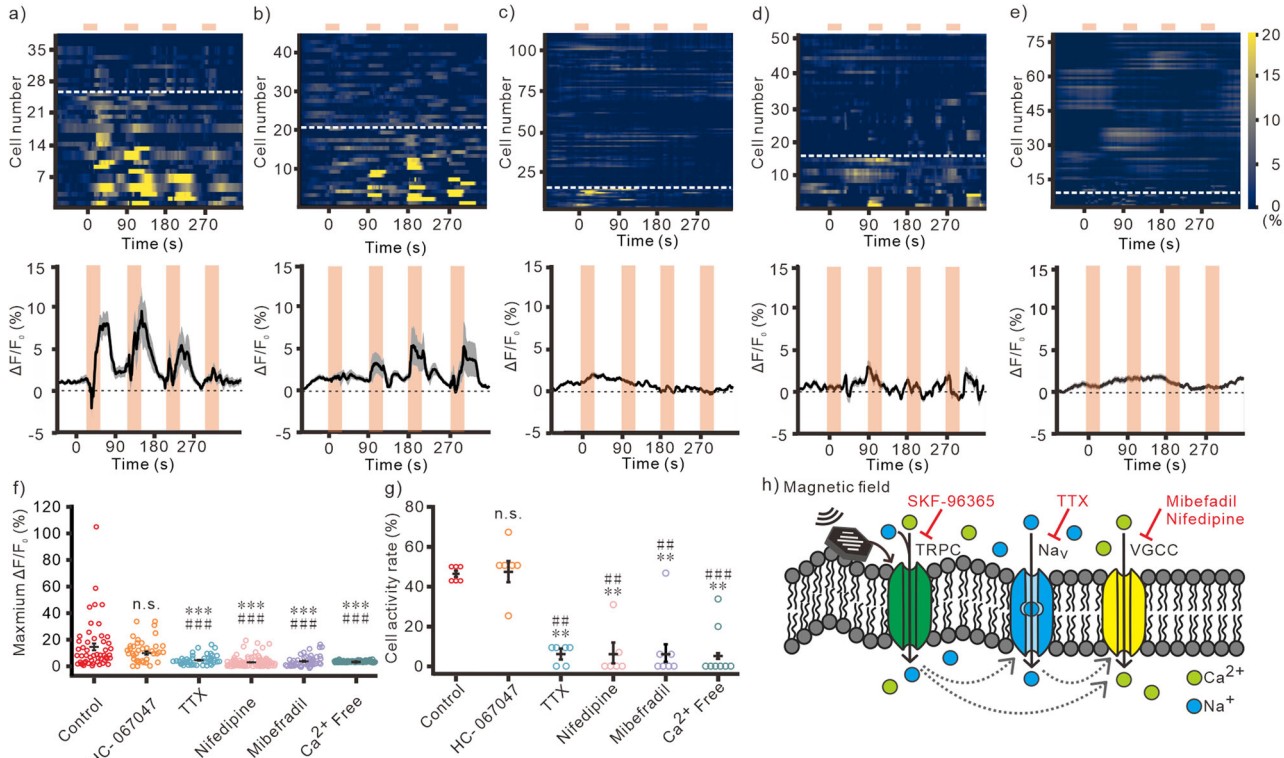

**Fig. 5 Pharmacological dissection of magnetomechanical stimulated responses in neurons.** The $Ca^{2+}$ responses by multiple magnetomechanical stimulation with bath application of HC-067047 at 1 μM (**a**, $n = 38/6$ (neurons/samples)), TTX at 100 nM (**b**, $n = 44/6$), Nifedipine at 10 μM (**c**, $n = 107/6$), Mibefradil at 3 μM (**d**, $n = 50/7$), and $Ca^{2+}$-free solution (**e**, $n = 79/8$). The magnetic stimulations are 50 mT at 10 Hz for 30 s. Inter-stimulation intervals are 60 s. top, heatmap of individual cells' responses. The cells below white dash lines were activated during stimulations. bottom, averaged fluorescence changes. The light orange areas indicate the time periods of magnetic stimulation. The gray areas are s.e.m. **f** The maximum fluorescence changes at the first magnetic stimulation (Control, $n = 52/6$ (neurons/samples); HC-067047, $n = 38/6$; TTX, $n = 44/6$; Nifedipine, $n = 107/6$; Mibefradil, $n = 50/7$; $Ca^{2+}$-free, $n = 79/8$). $F = 135.911$, $p < 0.001$, Kruskal-Wallis test. **g** The cell activity rate at the first magnetic stimulation (Control, $n = 6$; HC-067047, $n = 6$; TTX, $n = 6$; Nifedipine, $n = 6$; Mibefradil, $n = 7$; $Ca^{2+}$-free, $n = 8$). $F = 30.674$, $p < 0.001$, Kruskal-Wallis test. **h** Schematic for the mechanism of MND mediated responses. $**p < 0.01$, $***p < 0.001$, compared to the control group; $##p < 0.01$, $###p < 0.001$, compared to HC-067047 group; Dunn post-hoc test. Error bars represent mean ± s.e.m.

nanodiscs will be necessary to increase the specific attachment of nanodiscs to the targeted TRPC channels or to the targeted neurons by modifying the surface of the nanodiscs.

Most of the recently developed untethered DBS methods, including optogenetics, chemogenetics, sonogenetics and magnetogenetics, are using genetic tools to overexpress exogenous genes in the target brain regions. However, the regulatory barriers of genetic methods that used in those methods limit the translational applications of those DBS approaches in human patients. The herein-demonstrated magnetomechanical DBS approach is a transgene-free approach that does not have the safety concerns of using viral vectors for gene delivery. This study not only reveals the activation of intrinsic TRPC by magnetomechanical stimulation, but also reveals that the mechanical force-induced activation of intrinsic TRPC could trigger action potentials and activate the VGCC in neurons (Figs. 4, 5). With TTX, the MNDs-induced responses were significantly reduced. And it indicates that MNDs can trigger action potentials in neurons. Interestingly, there were remaining TTX-independent $Ca^{2+}$ responses when neurons were stimulated by multiple magnetomechanical stimulation. Which might recruit more $Ca^{2+}$ permeable ion channels. Low-threshold VGCC, including T-type VGCC and $Ca_V1.3$ of L-type VGCC, are highly expressed at the soma and dendrite of hippocampal neurons[40,41]. By using VGCC blockers, we found that these remaining magnetomechanically-induced $Ca^{2+}$ responses were contributed by activation of T-type and L-type

VGCC. However, we didn't observe an obvious $Ca^{2+}$ influx via TRPC. Although TRPC families are cation channels with calcium permeability. The $Ca^{2+}$ permeability depends on the subunit composition of the TRPC heteromer. The $Ca^{2+}$ permeability of heteromeric TRPC, like TRPC1/5 and TRPC1/6, can be reduced by TRPC1 subunit[42,43]. Those heteromeric TRPC with TRPC1, 5, and 6 subunits are highly expressed in hippocampus and STN. Further investigation is required to understand which TRPC subunit compositions are related to magnetomechanical stimulation. The understanding of the TRPC subunits expression in the target brain regions and target cell-types is also crucial for future applications.

In Parkinson's diseases, DBS at STN can rescue the abnormal behaviors[1]. However, in healthy WT mice, the relation between animal behavior and DBS at STN are unclear. A study shows that activation of STN by optogenetics can trigger ipsilateral rotation in WT mice[44]. In contrast, another study shows that stimulating STN with magnetothermogenetics increases contralateral rotation in WT mice[13]. Those studies demonstrate very controversial results of DBS at STN in mice without Parkinson's disease. The difference of these controversial findings might relate to the neuromodulation methods and stimulation intensities. In this study, the rotation behavior of MNDs injected mice are not changed by MND-mediated magnetomechanical DBS at STN in awake mice. Nevertheless, with immunostaining of c-fos, we found the increasement of neuronal activity in the WT mice

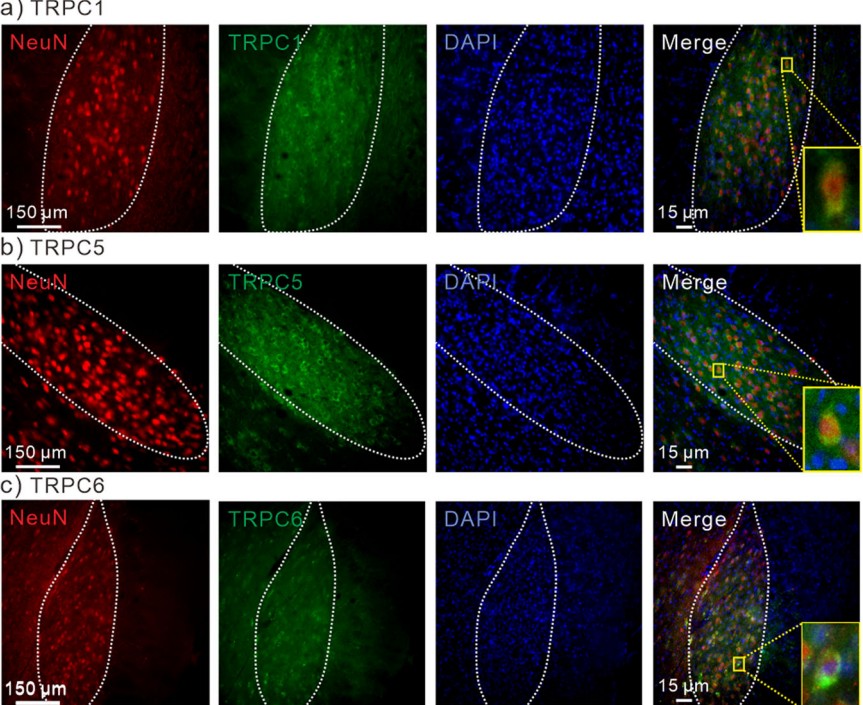

**Fig. 6 TRPC1, 5 and 6 expression in STN of mice. a** Immunostaining of NeuN (red), TRPC1 (green), DAPI (blue) and merged image in STN. Inset, enlarged merged image of a neuron. **b** Immunostaining of NeuN (red), TRPC5 (green), DAPI (blue) and merged image in STN. Inset, enlarged merged image of a neuron. **c** Immunostaining of NeuN (red), TRPC6 (green), DAPI (blue) and merged image in STN. Inset, enlarged merged image of a neuron.

with wireless magnetomechanical DBS. Our research demonstrates a proof-of-concept for wireless stimulating neuronal activity in vivo. Further investigation is required to optimize the parameters of magnetomechanical stimulations in vivo. Regulating the TRPCs are previously proposed for treating Parkinson's diseases[25] and ischemic stroke[27]. Unfortunately, the functional roles of TRPCs in most brain regions haven't been well characterized. The differences of TRPC functional roles in various brain regions and cell types must be considered for future applications. Further translational study will be necessary to find out the potential application of magnetomechanical stimulation in Parkinson's diseases and other neurological diseases.

Other than genetic-based neuromodulation approaches, some transgene-free wireless-powered miniature DBS devices still require millimeter to centimeter-scale hardware implantations into the deep brain regions[45,46]. In nanoscale transgene-free neuronal modulation, a study shows that magnetic stimulation with 200 mT DC magnetic field and 6 mT AC magnetic field at 140 Hz can activate cobalt ferrite-barium titanate piezoelectric nanoparticles for wireless DBS in freely moving mice in vivo[47]. In comparison, the magnetomechanical DBS in this study only requires the weak magnetic field with 50 mT at very low frequency (10 Hz). Thus, the magnetic apparatus for generating homogeneous magnetic fields in range for magnetomechanical stimulation is scalable to larger volumes. The custom-made coil for in vitro and in vivo experiments in this study had 3.5 cm and 20 cm inner diameter, respectively (Figs. 2c, 7b). It can be incorporated into most microscope systems and animal behavioral apparatuses. The scalable feature of this approach is ideal for future applications in larger animal models or humans. In addition, iron oxide magnetic nanoparticles are already clinically approved as contrast agents in magnetic resonance imaging (MRI)[6]. The iron oxide magnetic nanodiscs in this study have similar chemistries to those clinically approved nanoparticles. Therefore, this work is an important proof-of-concept

in transgene-free remote DBS with magnetomechanical stimulation, which shows promise for developing future translational applications for human patients.

## Methods

**Hematite and magnetite nanodiscs synthesis**. Synthesis process of nanodiscs follows the protocol described in the previous study[11]. The synthesis process consists of two steps. First, hematite nanodiscs were synthesized by mixing 10 ml 99.5% ethanol, 0.6 ml ddH$_2$O (or 0.8 ml ddH$_2$O for smaller nanodiscs), 0.8 g of anhydrous sodium acetate (Sigma-Aldrich) and 0.273 g of FeCl$_3$·6H$_2$O (Sigma-Aldrich). After the mixture was homogenized by stirring, the mixture was transferred and sealed into a Teflon-lined steel vessel. The vessel was heated in the oven at 180 °C for 18 h. The hematite nanodiscs were washed with ddH$_2$O twice. Then nanodiscs were washed with ethanol twice. Nanodiscs were dried in a vacuum desiccator. Hematite nanodiscs were further converted into magnetite nanodiscs or used for control experiments. For reduction, 1 mg hematite nanodiscs were mixed with 20 ml of tri-octylamine (Acros Organics) and 1 g of oleic acid (Sigma-Aldrich). The mixture was placed in a three-neck flask connected to a Schlenk line to heated to 370 °C for 25 min in an atmosphere of H$_2$ (5% with 95% Argon, Chiah Lung) and N$_2$ (99.9%, Chiah Lung). During reduction, the red hematite nanodiscs turned to dark gray. After cooled down, the discs were washed with hexane (Alfa Aesar). The magnetite nanodiscs were then dispersed in chloroform (J.T. Baker) and stored in a glass vial at 4 °C.

**Surface functionalization and water transfer**. Both MND and HND are functionalized with PMAO-coating. Before PMAO coating, nanodiscs in chloroform were dried in a vacuum desiccator. First, 10 mg 30,000 mw PMAO (419117, Sigma-Aldrich) was dissolved in 1 ml chloroform. Dried 1 mg MND or HND powder were added to the PMAO-containing chloroform. The mixture was sonicated for 1 h until the nanodiscs were well suspended. Next, the PMAO-coated nanodiscs were dried in a vacuum desiccator overnight. The dried nanodiscs were added into the TAE buffer (Tris-acetate-EDTA, Biomate) with concentration at 25% (w/v). The mixture was sonicated at 80 °C for 3 h. After sonication, it was pelleted by using a microcentrifuge for 10 min at 8500 rpm. Removed the supernatant and refilled with ddH$_2$O. Repeated the processes from well-sonication to ddH2O-refill for three times. For visualizing nanodiscs in the brain slices, Fluorescence-labeled PMAO-coated MNDs were prepared by mixing 200 μl Neutravidin (Thermo Scientific) with 200 μl of Alexa Fluor 594 dye (Thermo Scientific) for two hours. Then added 10 mg MNDs to Neutravidin-dye mixture for two hours.

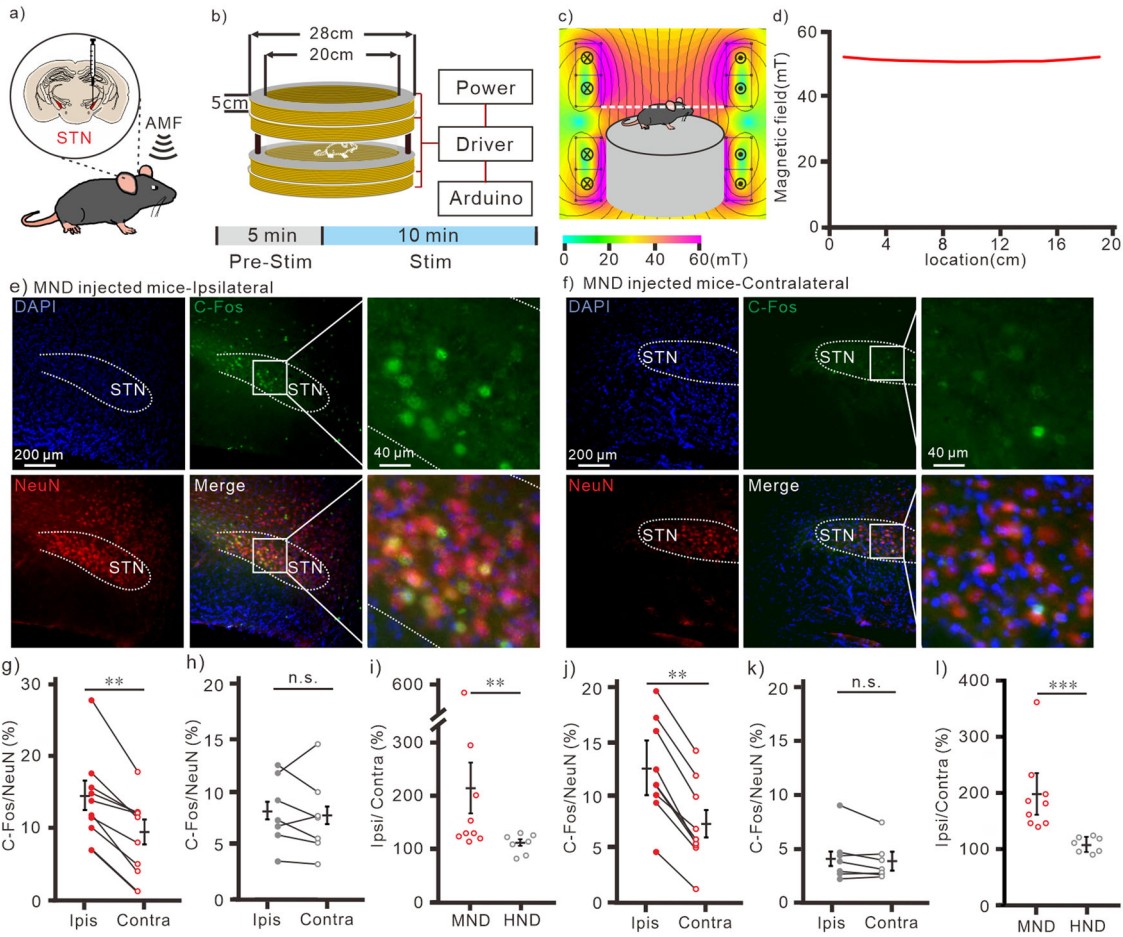

**Fig. 7 magnetomechanical stimulation in awake mice in vivo. a** Schematic of magnetomechanical stimulation at STN in vivo. Nanodiscs were unilaterally injected into STN. bottom, Timeline of stereotaxic injection, magnetic stimulation and c-fos staining **b** Schematic of wireless magnetic stimulation apparatus for magnetomechanical stimulation in vivo. bottom, Timeline for magnetic stimulation. **c** FEMM simulation of the in vivo magnetic apparatus with 10 A current. The black line indicates the cylinder chamber for mice. **d** Measurement of magnetic field from the white dash line indicates region in **c**. Immunostaining of NeuN (red), c-fos (green), DAPI (blue), and merged image from ipsilateral STN (**e**) and contralateral STN (**f**) of MND injected mice. **g** c-fos/NeuN of STN in MND injected mice ($n = 9$). **h** c-fos/NeuN of STN in HND-injected mice ($n = 7$). **i** The difference of c-fos/NeuN between ipsilateral and contralateral STN in nanodiscs injected mice. **j** c-fos/NeuN of EP in MND-injected mice ($n = 9$). **k** c-fos/NeuN of EP in HND-injected mice ($n = 7$). **l** The difference of c-fos/NeuN between ipsilateral and contralateral EP in nanodiscs injected mice. **p < 0.01, ***p < 0.001, n.s., no significance. Wilcoxon signed-rank test for paired data in (**g**), (**h**), (**j**) and (**k**). Mann-Whitney test for unpaired data in **i** and **l**. Error bars represent mean ± s.e.m.

**Nanodiscs characterization**. Nanodiscs in chloroform were dried in a vacuum desiccator and then dissolved in ddH$_2$O. The aqueous nanodiscs were placed on a copper grid (Ted Pella Inc.) and used transmission electron microscopy (TEM) for visualizing the morphology of the discs. TEM was performed with HT7800 (Hitachi) by the microscopy core laboratory at Chang Gung Memorial Hospital (CGMH). Diameter and thickness of nanodiscs were measured by ImageJ from TEM image, each hexagonal nanodiscs were drawn three diagonals and chose the longest distance as one nanodisc diameter measurement. Nanodisc powders were detected by X-ray diffraction (XRD) to confirm the structure of materials. XRD was performed with XtaLAB Synergy DW (Rigaku) by the Instrumentation Center at National Tsing Hua University (NTHU). Hematite or magnetite were mixed with N-grease (Apiezon) and collected on MicroMeshes$^{TM}$ (MiTeGen) coupled to HyPix-Arc 150° curved Hybrid Photon Counting X-ray detector with graphite-monochromated Mo Kα radiation ($\lambda = 0.71073$ Å) at 100 K. For studying saturation magnetization (Ms) of nanodiscs, vibrating sample magnetometer was used to measure hysteresis curves in the range of ± 9 kOe. Magnetic moments of nanodiscs were measured with MPMS SQUID Vibrating Sample Magnetometer (VSM; Quantum Design) by the Core Facility Center of National Cheng Kung University (NCKU). Further, an Agilent 725 Inductively Coupled Plasma Optical Emission Spectrometer (ICP-OES) was used to quantify the iron element concentration for the calculation of magnetic moment which was performed by the Instrumentation Center at NTHU. For calculating magnetic moment, we followed the formula mentioned in previous study[11]:

$$|\mu| = V \cdot \rho \cdot M_s \qquad (1)$$

μ is the magnetic moment. V is the volume of nanodiscs, which was measured from TEM. ρ is the density of magnetite (Fe$_3$O$_4$)[11]. M$_s$ is the saturation magnetization that is measured from VSM. The magnetic moment of MNDs is:

$$1.97 \times 10^{-21} \, m^3 ((5150 kg/m^3)((86 \, Am^2)/kg) = 8.7 \times 10^{-16} Am^2 \qquad (2)$$

PMAO-coated nanodiscs dissolved in ddH$_2$O were used for detecting potential properties. Zeta potential was measured by electrophoretic light scattering with Delsa$^{TM}$ Nano C Particle Analyzer (Beckman Coulter). 3.5 μl of 20 mg/ml PMAO-coating nanodiscs were loaded on hippocampal neurons at 7 days in vitro (DIV) for 15 min to attach nanodiscs on cell membranes. Then use 1× PBS (Gibco) to wash three times with the fixation buffer made by CGMH (3% glutaraldehyde, 2% paraformaldehyde in 0.1 M cacodylate buffer). Fixed samples were stored in 4 °C freezer until using field emission scanning electron microscopy (FE-SEM) to record nanodiscs attaching on neurons. FE-SEM was performed with SU8220 (Hitachi) by the microscopy core laboratory at CGMH. FE-SEM was used to record nanodiscs attaching on neurons at 2000× magnification and the surface features of the nanodiscs for 30,000× magnification.

**Primary hippocampal neuronal culture**. All animal experimental procedures were approved by the Institutional Animal Care and Use Committee (IACUC) of National Yang Ming Chiao Tung University (NYCU). All pregnant Sprague-Dawley rats were perchased from LASCO. The pups of Sprague-Dawley rats at postnatal (<3 days) were used for primary hippocampal culture. Hippocampi were extracted in cold dissection solution (160 mM NaCl, 5 mM KCl, 1 mM MgSO$_4$, 4 mM CaCl$_2$, 5 mM HEPES, 5.5 mM glucose, pH was adjusted to 7.4 by NaOH). After extraction, hippocampi from 2 to 3 pups were mixed and transferred into prewarmed digestion solution (1 mM L-Cysteine, 0.5 mM EDTA, 1 mM CaCl$_2$, 1.5 mM NaOH, and 10 units/ml papain (76220, Sigma-Aldrich)). The mixture was

incubated at 37 °C in for 25 min. Papain was inactivated by removing the digestion solution and incubating the tissues in inactivation solutions (0.25% bovine albumin and 0.25% trypsin inhibitor), 0.4% D-glucose, and 5% fetal bovine serum (6140079, Gibco) in Minimum Essential Medium (MEM w/Earle's salts w/o L-glutamine; 11090-081, Gibco) at 37 °C for 2 min. After removing the inactivation solution, tissues were triturated in serum medium (0.4% D-glucose, and 5% fetal bovine serum in MEM w/Earle's salts w/o L-glutamine) with fire-polished glass pipetted (111096, Kimble). The dissociated cells were filtered by cell-drainer (93070, SPL) and incubated in the serum medium at 37 °C before seeding. After counting, cells were seeded on the matrigel (354234, Corning) coated 12 mm coverslips in 24-well plates with ~110,000 cells per well. Hippocampal cells were cultured in neurobasal medium (10888-022, Gibco) with B27 supplement (17504-044, Gibco) and GlutaMAX (35050-061, Gibco). On 3rd days in vitro (DIV) 20 µl mitotic inhibitor (5-fluoro-2′-deoxyuridine, Sigma; 4 µM in neurobasal medium) was added to inhibit glial cells. All the imaging and stimulation are performed at DIV 5-14.

**Patch-clamp electrophysiology.** Primary cultured hippocampal neurons at DIV 7-14 were used for patch-clamp electrophysiological recording. The whole-cell recordings were obtained using MultiClamp 700B (Molecular Devices, LLC, USA) under an up-right microscope (Scientifica, EN). Data were filtered at 5 kHz and sampled at 10 kHz with a Digidata 1550B interface (Molecular Devices) controlled by Clampex 11.1 software (Molecular Devices) and analyzed using Clampfit 11.2 (Molecular Devices). Recordings were made at room temperature. The glass pipettes with pipette resistance at 3 to 10 MΩ were pulled from borosilicate glass (Harvard Apparatus, USA). The internal solution contained 74 mM KCl, 70 mM K-gluconate, 0.2 mM EGTA, 4 mM MgATP, 10 mM HPEPS and 7 mM Na₂-phosphocreatine, adjusted to pH 7.3 by KOH. The extracellular Tyrode's solution contained with 125 mM NaCl, 2 mM KCl, 2 mM MgCl₂, 2 mM CaCl₂, 25 mM HPEPS and 51 mM D-glucose, adjusted to pH 7.3 by NaOH. After break-in and form the whole-cell recording, the resting membrane potentials of each neuron were measured immediately. The seal resistances of recording were 3 to 10 MΩ. Rheobase was the minimum 1 s current step that could trigger more than one action potential. Input resistances were recorded with −50 pA current step for 1 s. The input resistances were measured from the voltage difference between baseline and the last 100 ms of current application.

**Ca²⁺ imaging with nanodiscs.** Tyrode's solution (125 mM NaCl, 2 mM KCl, 2 mM MgCl₂, 2 mM CaCl₂, 25 mM HEPES, 51 mM D-glucose (Sigma-Aldrich)) was prepared for all Ca²⁺ imaging for cultured cells. Fluo-4 Ca²⁺ Imaging Kit (Invitrogen) was used for measuring the Ca²⁺ responses of cultured neurons. The protocol for fluo-4 imaging was indicated by Invitrogen. The neurons were incubated in the Fluo-4 solution (1 mM) for 15 to 30 min, and then were transferred to Tyrode's solution without Fluo-4 for imaging. 3.5 µl magnetite nanodiscs or hematite nanodiscs at a concentration of 20 mg/mL was added to each well with 496.5 µL solution in 24-wells plate. The final concentration of nanodiscs is 70 µg/well. After 5 min incubation, the coverslip with hippocampal neuron was transferred to a custom stage for applying magnetic field under fluorescence microscope (SS-1000-00, Scientifica). Thorlabs light source (LEDD1B), Hamamatus C13440 camera, GFP filter cube (39002, Chroma) was used for fluo-4 imaging.

**Ca²⁺ imaging analysis.** Ca²⁺ activity videos were collected on an upright fluorescence microscope (SS-1000-00, Scientifica) in the ".avi" format using the HCImage software (Hamamatsu). Video's frame rate was 1 Hz. The videos were processed with a custom python script based on opencv2. A custom python script based on numpy was used for converting fluorescence intensity into ΔF/F₀ by adapting the algorithm from previous study[11]. The detail of algorithm and python script is described in supplementary information. The activated cell was defined by the cell with maximum ΔF/F₀ more than 10% at indicated time periods. The cell activity rate of each culture sample was defined by the percentage of activated cell number divided by the total cell number. For more details of the custom scripts, see the supplementary information.

**Pharmacology experiments.** The Tyrode's solution without Ca²⁺ (125 mM NaCl, 2 mM KCl, 2 mM MgCl₂, 2.5 mM EGTA, 25 mM HEPES, 51 mM D-glucose (Sigma-Aldrich)) was prepared for Ca²⁺ free experiments. The TRPC family inhibitor SKF-96365 (Tocris) was added to Tyrode's solution with Ca²⁺ at a concentration of 50 µM. The TRPV4 specific inhibitor HC-067047 (Sigma) was added to Tyrode's solution at a concentration of 1 µM. The Tetrodotoxin (TTX) sodium channel blocker (Abcam) was added to Tyrode's solution at a concentration of 100 nM. The TRPC 1/6 and piezo1 specific inhibitor GsMTx4 (Abcam) was added to Tyrode's solution at a concentration of 5 µM. The TRPC 1/6 and piezo2 specific inhibitor D-GsMTx4 (Tocris) was added to Tyrode's solution at a concentration of 5 µM. The TRP inhibitor 2-APB (Tocris) was added to Tyrode's solution at a concentration of 100 µM. In pharmacology experiments, cultured cells were transferred from Tyrode's solution with Fluo-4 to above-mentioned solutions for more than 5 min before imaging.

**Animal experiments and Stereotaxic injection.** All the animal experiments were approved by NYCU IACUC, in accordance with the Guide for the Care and Use of

Laboratory Animals of NYCU. All mice were purchased from LASCO. Mice were maintained under a 12 h light-dark cycle at NYCU Laboratory Animal Center before experiments. 8 to 12 weeks old C57BL/6 male mice were used in all in vivo experiments. PMAO-coated magnetite and hematite nanodiscs at 1 mg/ml were unilaterally injected into STN (AP: −2.06, ML: -1.5, DV: −4.5). Total 2 µl of nanodiscs in PBS were injected by using a microinjection syringe (7803-05, Hamilton) and a micropump (Kd Scientific). After cranial surgery, 0.2 ml of Carprofen (PHR1452, Sigma-Aldrich) was given to mice through subcutaneous injection at 0, 24, 48 h for reducing postsurgical pain.

**Fixed brain slicing and immunostaining.** For c-fos staining, nanodiscs injected C57BL/6 male mice were placed in a custom-made coil for magnetic stimulation in vivo 5 to 7 days after nanodiscs injection. Mice were sacrificed 90 min after magnetic stimulations. For staining TRPC, C57BL/6 male mice without magnetic stimulation were used. For staining fluorescence-labeled MNDs, C57BL/6 male mice were sacrificed at the same day or 5 days after fluorescence-labeled MNDs were inject at STN. All brains were collected following transcranial perfusion with 4% PFA in PBS. Coronal brain sections were sliced by a vibratome (5100 MZ, Campden) with amplitude 0.5, frequency 50 Hz. Slices for immunostaining had 50 µm thickness. Slices with fluorescence-labeled MNDs had 150 µm thickness. The brain slices were washed three times with PBS for 5 min, the slices were permeabilized with 2% (v/v) Triton X-100 (Sigma-Aldrich) for 15 min. Background was cleared up with 2% Triton-X-100, 30% H₂O₂, and methanol for 10 min; After the slices were washed by PBS for three times, the slices were blocked by 3% normal goat serum in PBS for 90 min at room temperature. Slices were washed with PBS ~~for~~ three times. For immunostaining, slices were incubated with 1st antibody solution with 1:750 rabbit anti-c-Fos monoclonal antibody (9F6#2250, Cell signaling), 1:150 mouse anti-NeuN antibody (clone A60, #MAB377, MERCK), 1:200 anti-rabbit NeuN (MABN140, Sigma-Aldrich), 1:100 anti-rabbit TRPC1 (SI-T8276, Sigma-Aldrich),1:500 anti-mouse TRPC5 (N67/15, NeuroMab), 1:100 anti-rabbit TRPC6 (AB5574, MERCK), 1% normal goat serum and 2% Triton-X 100 in PBS. Slices were incubated at 4 °C for 16–18 h. After three times washes of the slices with PBS, slices were incubated with matching secondary antibody in the PBS (1:500 goats anti-rabbit Alexa Fluor 488 (ab150113, Abcam) and 1:500 goat anti-mouse Alexa Fluor 594 (ab150116, Abcam)). All slices were washed with PBS ~~for~~ three times before mounting. Slices were mounted on glass microscope slides by mounting medium with DAPI (GTX30920, Genetex). Inverted fluorescence microscope (DMI3000, Leica), upright fluorescence microscope (SS-1000-00, Scientifica), LED light source (pE300, CoolLED), Hamamatsu C13440 camera, filter cubes (39000, 19008, 31002, Chroma) was used for imaging.

**Magnetic apparatus for fluorescence microscope.** The coil was an air-core coil made by 2000 turns of 18 AWG self-bonding copper wire (SBWR, Chientai). The coil had 7 Ω Resistance and 60 mH inductance. At 1 Hz varying magnetic field stimulation, 0.6 A to 2.8 A was used for 10 to 50 mT stimulation. At 5 Hz magnetic stimulation, 0.6 A to 2.9 A was used for 10 to 50 mT stimulation. At 10 Hz magnetic stimulation, 0.6 A to 3 A was used for 10 to 50 mT stimulation. At 20 Hz magnetic stimulation, 0.7 A to 3.2 A was used for 10 to 50 mT stimulation. A custom-made H-bridge driver was used for generating varying magnetic fields with coil. The H-bridge was made up of two p-channel MOSFET (IRF4905, International Rectifier) and two n-channel MOSFETs (IRF3710, International Rectifier). Beside these four MOSFETs, two more n-channel MOSFETs were used to control P-channel MOSFETs that in H-bridge. Range of working voltage is controlled by two pairs of resistances and parameters of MOSFET. Circuit can easily be modified by replacing different resistance or MOSFET. Function generator (33210 A, Keysight) was used for generating ± 5 V square waves. The signal from the function generator passed through a voltage follower and an inverter (TLC2272, Texas Instrument). The gains of voltage follower and inverter equal to one. The phases of signals from voltage follower and inverter have 180° difference. Each signal controls a half-bridge of the full-bridge driver. Internal power supply (PS-3030DF model, LONGWEI) was used for op amps on the custom-made H-bridge driver. External power supply (IT6721, ITECH) was used for generating magnetic fields in the coil.

**Magnetics simulation and magnetic field measurement.** To evaluate magnetic fields generated by coils, we use Finite Element Method Magnetics software (version 4.2) to simulate magnetic fields of coil in 2D (Figs. 2d, 5c). In program setting, we used magnetics problem mode and planar configuration in FEMM4.2. The parameters of copper wire (12 or 18 AWG) and air were from the materials library in FEMM4.2. Gauss meter (TM801, KANETEC) was used for magnetic field measurement in the coil. The axial probe was used to detect the magnetic field that generated inside of the coil.

**Cell viability test.** The death of primary cultured neurons with multiple magnetomechanical stimulations were measured with propidium Iodide (PI; P1304MP, Invitrogen). After imaging the Ca²⁺ responses of primary cultured neurons with Fluo-4, PI was added to the extracellular solution to achieve a final concentration at 120 µM. After 5 min, the fluorescence of Fluo-4 and PI were imaged by using fluorescence microscope (SS-1000-00, Scientifica). Thorlabs light source

(LEDD1B), Hamamatus C13440 camera, and filter cubes (39002 and 39010, Chroma) were used.

**Magnetic apparatus for in vivo experiments**. To generate a uniform magnetic field for wireless neuronal stimulation in vivo. Four custom-made air-core coils were used for in vivo experiments. Each coil has 500 turns of 12 AWG copper wire. The resistance and inductance of each coil is 1.02 to 1.55 Ω and 22 to 31.6 mH, respectively. As demonstrated in Fig. 6, four coils are separated into two pairs which stack on each other. The gap between coils is 4 cm. Full bridge modules (AQMH3615NS, AKELC) were used as the drivers for each coil. The drivers were controlled by the 5 V square waves generated by Arduino UNO (Arduino). The Arduino board was controlled by a custom-made script with Arduino IDE. The detail of the Arduino code was described in the supplementary information. Internal power supply is 5 V and provided by Arduino to give working voltage for full bridge modules. External power supply (HJS-1000 model, HuntKey) was used for supplying the currents for the coils. 10 A currents were used for generating a uniform magnetic field with 50 mT at 10 Hz in the coils for in vivo experiments.

**Magnetic stimulation in vivo and rotation behavioral test**. All the animals habituated in the behavior room for at least 45 min before magnetic stimulation. The cylinder arena was made out of Polymethylmethacrylate with 20 cm inner-diameter and 16 cm height, fitting within the custom-made magnetic apparatus. The awake mice were placed into the cylinder arena in the center of the coil and recorded by camera from top. Before stimulation, there is 5 min of pre-stimulation period without applying magnetic fields, followed by 10 min of magnetic stimulation period. In the magnetic stimulation period, 30 s stimulation by alternative magnetic field (50 mT at 10 Hz) and 30 s inter-stimulation intervals without magnetic field were applied for 10 times. This magnetic stimulation process was used for investigating neuronal activity with c-fos and for rotation behavioral tests. The previously reported DeepLabCut[TM] (DLC)[48] was used for tracking markless mice in the recorded behavioral videos. A custom-made python code was used for analyzing the rotation behaviors. Please see the supplementary methods for detailed description and scripts.

**Statistics and reproducibility**. All statistics were performed in JASP (v0.14.1.0, JASP team). All the error bars in the dot plots indicate standard error of mean (s.e.m.). All the gray or pink areas in the traces of fluorescence changes indicate standard error of mean (s.e.m.). Wilcoxon signed-rank test was used for comparing paired data. Wilcoxon rank-sum test was used for comparing unpaired. Tukey post-hoc test and Two-way ANOVA was used for comparing data with two factors. Dunn post-hoc test and Kruskal-Wallis test were used for comparing data with multiple groups. The numbers of individual cells, samples (individual cultures), and animals of each experiments are indicated in the legends of figures and tables

**Reporting summary**. Further information on research design is available in the Nature Research Reporting Summary linked to this article.

## Data availability

Source data underlying figures used in the current study are provided in Supplementary Data 1. All other data are available from the corresponding authors upon reasonable request.

## Code availability

The code related to this study is available in Methods section of supplementary information.

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

## Acknowledgements
We thank P. Ankieeva, D. Gregurec, and F. Koehler for materials and electronics advice. This research is funded by Ministry of Science and Technology (MOST), Taiwan (R.O.C.) (108-2636-B-009-006; 109-2636-B-009-008; 110-2636-B-A49-003).

## Author contributions
Conceptualization: P.H.C., C.L.S. Methodology: P.H.C., C.L.S., C.C.C., P.H.Y., J.X.H. Investigation: C.L.S., C.C.C., P.H.Y., J.X.H., Y.J.T. Formal analysis: C.L.S., C.C.C., J.X.H., Y.J.T. Visualization: C.L.S., C.C.C., P.H.Y., J.X.H., Y.J.T., P.H.C. Funding acquisition: P.H.C. Project administration: P.H.C. Supervision: P.H.C. Writing—original draft: P.H.C. Writing—review & editing: P.H.C., C.L.S., C.C.C., P.H.Y., J.X.H., Y.J.T.

## Competing interests
The authors declare no competing interests.
