## [Peer Review File · Communications Biology]

Reviewers' comments:

Reviewer #1 (Remarks to the Author):

This study by Su et al utilizes magnetomechanical stimulation with magnetic MDN and HND nanodiscs to demonstrate activation of neurons via endogenous TRPC channels, and applies this approach for deep brain stimulation of the STN in vivo. Overall, this is an exciting study and a potentially important advance in the field. That said there are major issues with data interpretation and some moderate weaknesses in experimental methodology that need to be addressed. These issues are listed below in a point-wise fashion:

(1) The authors interpret their data as follows: "activation of TRPC by MNDs depolarized the membrane to induce action potentials". A major issue with this interpretation is that MNDs are negatively charged and are adhered to the outer surface of the neuron. This would cause the neuron to become depolarized because of excessive negative charge on the outer membrane of the neuron. This idea has not been considered. Because of MND induced depolarization, the function of voltage gated sodium channels in inducing neuronal depolarization has been bypassed. This notion agrees with the author's finding that TTX causes only a 50% reduction in calcium event amplitude in the neurons. It is the depolarization of neurons by MNDs that is causing calcium influx into the neurons, and not the other way around. This important issue needs to be carefully considered and the resting membrane potential of the cultured neurons with and without MNDs needs to be compared using either electrophysiology or optogenetic voltage sensors.

(2) Because of the idea that neurons may be depolarized by MNDs, it is likely that a proportion of the calcium fluxes observed with MND + magnetic fields are mediated by voltage gated calcium channels (VGCCs). This needs to be tested in the presence of VGCC blockers such as nifedipine for L-type VGCCs and mibefradil for T-type VGCCs.

(3) In the in vivo experiment, did the authors observe rotational behavior in the mice with unilateral injection of MNDs into the STN? This would be expected if there is unbalanced dopamine release into the striatum following MND injection into the STN and magnetic stimulation. The authors need to acquire videos of the mice under magnetic stimulation if they are not anesthetized and record rotational behavior.

(4) In the in vivo experiment, c-fos expression seems to be restricted in the images to the ventromedial aspect of the STN. It is unclear if this is a true biological phenomenon or if it is because of only a partial injection of the STN using stereotaxic surgery. This point needs to be clarified by injecting fluorescently tagged MNDs into the STN in order to demonstrate clearly that the STN is accurately targeted by MNDs. Also, the methods do not state if in vivo sections were in the coronal, horizontal or sagittal plane. Please clarify this point.

(5) The discussion is very general and does not interpret the data in the paper in the context of what is already known in the literature. The discussion needs to be completely rewritten in the form of interpreting the author's data and should not be a general discussion of the field as it is now. Specifically, the issue of negatively charged MNDs causing depolarization of neurons and the downstream consequences of this needs to be carefully discussed.

(6) There are several grammatical and syntax errors in the writing. Please proofread the manuscript carefully

(7) Fig 1f and 1g: Please statistically compare the groups and give p values

(8) Fig 2m: The 5 Hz stimulation group population data does not match with the representative trace shown in Fig 2j. The trace shows a much smaller amplitude than the population data. Please either recheck and correct Fig 2m or use a more representative trace in Fig 2j that reflects the population data.

(9) Please give a detailed description of number of samples. For neurons, how many neurons per experiment, and from how many independent cultures for each experiment. For mice, how many mice per experiment and are these male or female mice or mixed?

Reviewer #2 (Remarks to the Author):

This paper is a proof-of-concept test to use iron oxide magnetic vortex nanodiscs (MNDs) for wireless stimulation of deep brain structures by targeting force-sensitive transient receptor potential canonical (TRPC) that also expresses in the CNS. It is shown that applying relatively weak magnetic fields (50mT at 10Hz) to neuronal cultures in MNSs induced enhancement of calcium signals, suggesting enhancement of neuronal activity. Further pharmacological experiments support that the signals can be blocked by TRPC antagonists, partially related to action potential firing, and dependent on calcium influx. They also injected the nanodiscs into the subthalamus nucleus and one of its targeted nucleus in mice in vivo and showed that magnetic stimulation induced increases in the number of c-Fos positive neurons in the target brain regions. They conclude that a magnetic nanodiscs-based magnetomechanical approach can be used for wireless neuronal stimulation in vitro and in awake mice in vivo.

Overall, the paper tested a novel hypothesis that MNDs may be used for DBS without the need of genetical modification of cells, which may have important translational significance. Although many mechanistic questions remain to be answered, the methods are solid and the results generally support their conclusions.

Specific comments:

1. The abstract could be more specific to provide readers with more information about the finding. For example, the statement “We further demonstrated the wireless DBS with magnetomechanical approach in awake mice in vivo” is too general.
2. It seems that hematite (α -Fe₂O₃) nanodiscs (HNDs) was used as a negative control throughout of the study, but . The authors should introduce its property in contrast to MND in the introduction or beginning of the Results, which support the result interpretation and conclusion.
3. N numbers in the in vitro and in vivo results should be reported in the paper. For example, in Figure 3 and 4, how many cultures/trials were the reported cell numbers from?
4. In the in vivo experiment, the stimulation was made 5-7 days after injection, which is a long time period. It is unclear whether the injected MNDs were still in the STN. The authors should try to show the location and distribution of the MNDs in the tissue (such as fluorescent dye labeling, EM).
5. The images of C-fos staining were low quality. The authors should provide higher magnification or better quality images to show the c-Fos and dual staining.

Dear Dr. Scemes and Editorial Board,

Thank you and referees for the positive feedback. Reviewer #1 gave us a very positive feedback and mentioned ***“This is an exciting study and a potentially important advance in the field.”*** Reviewer #2 also gave us a positive response and mentioned ***“The paper tested a novel hypothesis that MNDs may be used for DBS without the need of genetical modification of cells, which may have important translational significance.”*** We have implemented their comments in the revised version of our manuscript (all changes are highlighted in gray). Please find below the specific responses to the comments:

Specific Responses to Reviewer #1:

Comment of Reviewer #1) This study by Su et al utilizes magnetomechanical stimulation with magnetic MDN and HND nanodiscs to demonstrate activation of neurons via endogenous TRPC channels, and applies this approach for deep brain stimulation of the STN in vivo. Overall, this is an exciting study and a potentially important advance in the field. That said there are major issues with data interpretation and some moderate weaknesses in experimental methodology that need to be addressed. These issues are listed below in a point-wise fashion:

Responses: We thank the reviewer for the very positive comment: ***“This is an exciting study and a potentially important advance in the field.”*** We thank the reviewer for comments and revise our manuscript accordingly. Please find below the point-by-point response to the concerns raised by the reviewer.

Reviewer #1, Comment 1) The authors interpret their data as follows: “activation of TRPC by MNDs depolarized the membrane to induce action potentials”. A major issue with this interpretation is that MNDs are negatively charged and are adhered to the outer surface of the neuron. This would cause the neuron to become depolarized because of excessive negative charge on the outer membrane of the neuron. This idea has not been considered. Because of MND induced depolarization, the function of voltage gated sodium channels in inducing neuronal depolarization has been bypassed. This notion agrees with the author's finding that TTX causes only a 50% reduction in calcium event amplitude in the neurons. It is the depolarization of neurons by MNDs that is causing calcium influx into the neurons, and not the other way around. This important issue needs to be carefully considered and the resting membrane potential of the cultured neurons with and without MNDs needs to be compared using either electrophysiology or optogenetic voltage sensors.

Response: We thank the reviewer for raising these issues of our study. We apology for the unclear description in the original sentence. In this study, MNDs were used as nanotransducers that can transform magnetic fields into mechanical force. We found that the torque from MNDs during the alternative magnetic field could activate intrinsic TRPC in neurons. The activation of TRPC could trigger action potentials. MNDs cannot directly activate the neurons without magnetic fields. To precisely describe our finding, we rephrased this sentence into ***“This result indicated that the magnetomechanical stimulated TRPC activation could induce action potentials in neurons.”*** (page 16, line 2 to line 3)

The second issue raised here was that the negatively-charged MNDs might depolarize the neurons and increase the neuronal activity. To investigate this possibility, we measured the

intrinsic electrophysiology properties of cultured neurons with whole-cell recording. We didn't observe any significant differences between the resting membrane potentials of neurons that were treated with negatively-charged MNDs, with negatively-charged HNDs and without any nanodiscs. This result was added to the main text: *“With whole-cell recording, we didn't observe any significant difference of the resting membrane potential between neurons with and without negative-charged MNDs and HNDs (Fig. S2a-d). There were also no significant differences in the other intrinsic properties between groups (Fig. S2e-f).”* (Page 14, line 3 to line 6; supplementary figure S2). Except the results from patch-clamp electrophysiology, we also didn't observe increase of neuron activity by negatively-charged HNDs. There are no Ca^{2+} responses in cultured hippocampal neurons with negatively-charged HNDs (Fig. 2e-h). In c-fos immunostaining, the neuronal activity in vivo in the STN and its downstream EP of unilateral HNDs injected mice had no differences between ipsilateral and contralateral hemisphere (Fig. 7h, k). These results showed that PMAO-coated nanodiscs did not increase neuronal activity without magnetomechanical stimulation. The discussion of possible influence of the surface charge is added to the discussion section: *“The MNDs and HNDs in this study were functionalized with PMAO, and were carried negative surface charges (Fig. 1i). Previous study shows that the negative-charged nanoparticles can facilitate the attachment of the particle on the excitable neuronal membrane but not on the non-excitable glial membrane^{11,30}. In contrast, the positive-charged or neutral nanoparticle cannot attach to the neuronal membranes³⁰. Although the negative-charged nanoparticles at surface of neuronal membrane might increase the excitability³⁰, we didn't observe significant increase of resting membrane potential and neuronal activity both in vitro and in vivo. First, there are no Ca^{2+} responses in cultured hippocampal neurons with negative-charged HNDs (Fig. 2e-h). With whole-cell recording, the resting membrane potentials of MNDs-treated and HNDs-treated neurons were not significantly higher than nanodiscs-free control groups (Fig. S2a-c). Finally, in c-fos immunostaining, the neuronal activity in vivo in the STN and its downstream EP of unilateral HNDs injected mice had no differences between ipsilateral and contralateral hemisphere (Fig. 7h, k). Nevertheless, further modification of the surface of nanodiscs will be necessary to increase the specific attachment of nanodiscs to the targeted TRPC channels or to the targeted neurons by modifying the surface of the nanodiscs.”* (Page 18, line 15 to line 29).

Reviewer #1, Comment 2) Because of the idea that neurons may be depolarized by MNDs, it is likely that a proportion of the calcium fluxes observed with MND + magnetic fields are mediated by voltage gated calcium channels (VGCCs). This needs to be tested in the presence of VGCC blockers such as nifedipine for L-type VGCCs and mibefradil for T-type VGCCs.

Response: We thank the reviewer for the suggestion to investigate the contribution of VGCC in the magnetomechanically-induced responses. With TTX application, the first magnetomechanical stimulation was almost abolished. However, there were some calcium responses remaining when we applied multiple magnetic stimulation. These calcium responses might be contributed by VGCC. When we applied nifedipine and mibefradil, all the magnetomechanically-induced calcium responses were inhibited. These results indicated that the magnetomechanically-induced TRPC activation could trigger VGCC. These descriptions of these results were added into the result section: *“The TTX-independent Ca^{2+} responses might be contributed by TRPC alone or by other voltage-gated Ca^{2+} channels (VGCC). Among all VGCC, T-type VGCC and Cav1.3 of L-type VGCC are low-threshold-activated ion channels. To investigate whether VGCC are involved in*

magnetomechanical stimulated responses, antagonists of L-type and T-type VGCC were applied during multiple magnetomechanical stimulations. Nifedipine and Mibefradil are commonly referred as specific L-type VGCC antagonist and specific T-type VGCC antagonist, respectively. However, accumulating evidences show that they are non-specific antagonists which can block both L-type and T-type VGCC³⁵. By using nifedipine (10 μ M)³⁶, there were almost no Ca^{2+} responses with multiple magnetic stimulations (Fig. 5c, f-g, S5c-d). Similarly, with mibefradil (3 μ M)³⁷ application, Ca^{2+} responses by multiple magnetic stimulation were significantly reduced (Fig. 5d, f-g, S5c-d). These results indicated that the magnetomechanically-induced TRPC activation could trigger VGCC. The TTX-independent Ca^{2+} responses might not be contributed by Ca^{2+} influx through TRPC along. Finally, with Ca^{2+} -free extracellular solution, all the magnetomechanically-induced Ca^{2+} responses were eliminated (Fig. 5e-g, S5c-d). These results indicated that these Ca^{2+} responses were contributed by Ca^{2+} influx from external solution (Fig. 5h). Overall, from the pharmacological study, we found that the torque of MND generated by alternative magnetic field can induce Ca^{2+} influx from external solution. Which were mainly caused by VGCCs and action potentials in cultured neurons. The SKF-96365, GsMTx4, d-GsMTx4 experiments shows that this magnetomechanical stimulated responses were mainly mediated by intrinsic TRPC (Fig. 4a-f, S5a-b).” (page 16, line 7 to line 26). We also discuss these results in the discussion section: “This study not only reveals the activation of intrinsic TRPC by magnetomechanical stimulation, but also reveals that the mechanical force-induced activation of intrinsic TRPC could trigger action potentials and activate the VGCC in neurons (Fig. 4, 5). With TTX, the MNDs-induced responses were significantly reduced. And it indicated that MNDs can trigger action potentials in neurons. Interestingly, there were remaining TTX-independent Ca^{2+} responses when neurons were stimulated by multiple magnetomechanical stimulation. Which might recruit more Ca^{2+} permeable ion channels. Low-threshold VGCC, including T-type VGCC and Cav1.3 of L-type VGCC, are highly expressed at the soma and dendrite of hippocampal neurons^{41,42}. By using VGCC blockers, we found that these remaining magnetomechanically-induced Ca^{2+} responses were contributed by activation of T-type and L-type VGCC.” (page 19, line 7 to line 16)

Reviewer #1, Comment 3) In the in vivo experiment, did the authors observe rotational behavior in the mice with unilateral injection of MNDs into the STN? This would be expected if there is unbalanced dopamine release into the striatum following MND injection into the STN and magnetic stimulation. The authors need to acquire videos of the mice under magnetic stimulation if they are not anesthetized and record rotational behavior.

Response: We thank the reviewer for the suggestion to look into the rotation behaviors in mice with magnetomechanical DBS. Although in unilateral Parkinson’s disease model mice, DBS at STN can rescue the rotational behavior of PD mouse model, the behavioral relation with DBS at STN in healthy WT mice is not well understood. A study shows that activation of STN by optogenetics could trigger ipsilateral rotation in WT mice (Guillaumin et al., 2021). In contrast, another study shows that stimulating STN with magnetothermogenetics increase contralateral rotation in WT mice (Hescham et al., 2021). The difference of these controversial findings might relate to the neuromodulation methods and stimulation frequency and intensities. Unfortunately, the rotation behavior of MNDs injected mice in this study are not changed by magnetomechanical DBS at STN. Nevertheless, with immunostaining of c-fos, we found the increasement of neuronal activity at STN and its downstream brain region in the WT mice with wireless magnetomechanical

DBS. Further investigation is required to optimize the parameters of magnetomechanical stimulations for behavioral analysis. The discussion of this important issue is added into the discussion section: *“In Parkinson’s diseases, DBS at STN can rescue the abnormal behaviors¹. However, in healthy WT mice, the relation between animal behavior and DBS at STN were unclear. A study shows that activation of STN by optogenetics could trigger ipsilateral rotation in WT mice⁴⁵. In contrast, another study shows that stimulating STN with magnetothermogenetics increase contralateral rotation in WT mice¹³. Those studies demonstrated very controversial results of DBS at STN in mice without Parkinson’s disease. The difference of these controversial findings might relate to the neuromodulation methods and stimulation intensities. In this study, the rotation behavior of MNDs injected mice are not changed by MND-mediated magnetomechanical DBS at STN in awake mice. Nevertheless, with immunostaining of c-fos, we found the increasement of neuronal activity in the WT mice with wireless magnetomechanical DBS. Our research demonstrated a proof-of-concept for wireless stimulating neuronal activity in vivo. Further investigation is required to optimize the parameters of magnetomechanical stimulations in vivo. Regulating the TRPCs were previously proposed for treating Parkinson’s diseases²⁵ and ischemic stroke²⁷. Unfortunately, the functional roles of TRPCs in most brain regions haven’t be well characterized. The differences of TRPC functional roles in various brain regions and cell types must be considered for future applications. Further translational study will be necessary to find out the potential application of magnetomechanical stimulation in Parkinson’s diseases and other neurological diseases.”* (Page 19, line 24 to page 20, line 10).

Reviewer #1, Comment 4) In the in vivo experiment, c-fos expression seems to be restricted in the images to the ventromedial aspect of the STN. It is unclear if this is a true biological phenomenon or if it is because of only a partial injection of the STN using stereotaxic surgery. This point needs to be clarified by injecting fluorescently tagged MNDs into the STN in order to demonstrate clearly that the STN is accurately targeted by MNDs. Also, the methods do not state if in vivo sections were in the coronal, horizontal or sagittal plane. Please clarify this point.

Response: We apology for the unclear images. There were variation of c-fos expression location in different brain slices. The increasement of c-fos might be located at the ventral, dorsal or center of STN. To avoid the misinterpretation, we replaced the images with more representative images which have c-fos expression at the center of STN (Fig. 7). In addition, to clarify the injection site, we used the fluorescence-labeled MNDs to visualize the location of MNDs in brain slices. With unilateral injection of fluorescence-labeled MNDs at STN, we can observe the fluorescence at STN. The description of this experiment was added to the result section: *“The coordinate of injection was confirmed by using fluorescence-labeled MNDs (Fig. S6a-b). After injection, the fluorescence-labeled MNDs were not degraded or metabolized for at least 5 days (Fig. S6c-d).”* (Page 17, line 3 to line 5; Fig S6). The brain slices were all coronal sections. The description is added in the *“Fixed brain slicing and immunostaining”* section in material and methods: *“Coronal brain sections were sliced by a vibratome (5100 MZ, Campden) with amplitude 0.5, frequency 50 Hz.”* (Page 9, line 12 to line 13)

Reviewer #1, Comment 5) The discussion is very general and does not interpret the data in the paper in the context of what is already known in the literature. The discussion needs to be completely rewritten in the form of interpreting the author's data and should not be a general

discussion of the field as it is now. Specifically, the issue of negatively charged MNDs causing depolarization of neurons and the downstream consequences of this needs to be carefully discussed.

Response: We thank the reviewer for the suggestion to rewrite the discussion for more interpretation of our data. The discussion section was rewritten to interpret our data and to compare with other studies (page 18 to 20). The issue of negatively charged MNDs was added to the discussion section: “*Although the negative-charged nanoparticles at surface of neuronal membrane might increase the excitability³⁰, we didn’t observe significant increasement of resting membrane potential and neuronal activity both in vitro and in vivo. First, there are no Ca²⁺ responses in cultured hippocampal neurons with negative-charged HNDs (Fig. 2e-h). With whole-cell recording, the resting membrane potentials of MNDs-treated and HNDs-treated neurons were not significantly higher than nanodiscs-free control groups (Fig. S2a-c). Finally, in c-fos immunostaining, the neuronal activity in vivo in the STN and its downstream EP of unilateral HNDs injected mice had no differences between ipsilateral and contralateral hemisphere (Fig. 7h, k). Nevertheless, further modification of the surface of nanodiscs will be necessary to increase the specific attachment of nanodiscs to the targeted TRPC channels or to the targeted neurons by modifying the surface of the nanodiscs.*” (page 18, line 19 to line 29).

Reviewer #1, Comment 6) There are several grammatical and syntax errors in the writing. Please proofread the manuscript carefully

Response: We thank the reviewer for pointing out our errors. In the revision, we rewritten the contents and corrected all the grammatical or syntax errors.

Reviewer #1, Comment 7) Fig 1f and 1g: Please statistically compare the groups and give p values

Response: We thank the reviewer for the suggestion. The statistical results and p values of Fig 1f and 1g were described in Table S1 to S3.

Reviewer #1, Comment 8) Fig 2m: The 5 Hz stimulation group population data does not match with the representative trace shown in Fig 2j. The trace shows a much smaller amplitude than the population data. Please either recheck and correct Fig 2m or use a more representative trace in Fig 2j that reflects the population data.

Response: We apologize for misplacing the 5Hz group data in the original figure 2j. We corrected the figure in the revised manuscript.

Reviewer #1, Comment 9) Please give a detailed description of number of samples. For neurons, how many neurons per experiment, and from how many independent cultures for each experiment. For mice, how many mice per experiment and are these male or female mice or mixed?

Response: We thank the reviewer for the suggestion. All the numbers of neurons, samples (individual cultures) were added into figure legends and the result section. We were using male mice for all the experiments. The numbers of mice in each experiment were described in figure legends. The strain and gender of mice were mentioned in the materials and methods.

Specific Responses to Reviewer #2:

Comment of Reviewer #2) This paper is a proof-of-concept test to use iron oxide magnetic vortex nanodiscs (MNDs) for wireless stimulation of deep brain structures by targeting force-sensitive transient receptor potential canonical (TRPC) that also expresses in the CNS. It is shown that applying relatively weak magnetic fields (50mT at 10Hz) to neuronal cultures in MNSs induced enhancement of calcium signals, suggesting enhancement of neuronal activity. Further pharmacological experiments support that the signals can be blocked by TRPC antagonists, partially related to action potential firing, and dependent on calcium influx. They also injected the nanodiscs into the subthalamus nucleus and one of its targeted nucleus in mice in vivo and showed that magnetic stimulation induced increases in the number of c-Fos positive neurons in the target brain regions. They conclude that a magnetic nanodiscs-based magnetomechanical approach can be used for wireless neuronal stimulation in vitro and in awake mice in vivo.

Overall, the paper tested a novel hypothesis that MNDs may be used for DBS without the need of genetical modification of cells, which may have important translational significance. Although many mechanistic questions remain to be answered, the methods are solid and the results generally support their conclusions.

Responses: We thank the reviewer for the very positive comments: ***“The paper tested a novel hypothesis that MNDs may be used for DBS without the need of genetical modification of cells, which may have important translational significance.”*** We thank the reviewer for comments and revise our manuscript accordingly. Please find below the point-by-point response to the concerns raised by the reviewer.

Reviewer #2, Comment 1) The abstract could be more specific to provide readers with more information about the finding. For example, the statement “We further demonstrated the wireless DBS with magnetomechanical approach in awake mice in vivo” is too general.

Response: We thank the reviewer for the suggestion about the abstract. We rewrote the abstract with more information about the finding. *“The immunostaining with c-fos showed the increasement of neuronal activity by wireless DBS with magnetomechanical approach in vivo.”*

Reviewer #2, Comment 2) It seems that hematite (α -Fe₂O₃) nanodiscs (HNDs) was used as a negative control throughout of the study, but the authors should introduce its property in contrast to MND in the introduction or beginning of the Results, which support the result interpretation and conclusion.

Response: We thank the reviewer for the suggestion about the introduction. The size and geometry of HNDs from the first step of the synthesis process was similar to the MNDs produced at the end of synthesis. With VSM measurement, the HNDs could not be magnetized by magnetic field applications. Thus, the PMAO-coated HNDs are the ideal control for our study. The description of HNDs were added in the result section: *“The diameters of HNDs were 282.8 ± 10.2 nm and 221.0 ± 4.2 nm with respectively 6 % and 8 % (v/v) H₂O in the first step reaction. The diameters of MNDs were 280.0 ± 5.9 nm and 212.4 ± 7.0 nm when respectively used 6 % and 8 % (v/v) H₂O in the first step reaction (Fig. 1f-g). The sizes and geometries of HNDs and MNDs from the same*

Response to reviewers

synthesis process were similar (Fig. 1f-g, Table S1-3). The X-ray diffraction spectra (XRD) showed that the HNDs from the first step were fully reduced into MNDs (Fig. 1h). Similarly, the magnetization curves showed that the HNDs cannot be magnetized with external magnetic field application (Fig. S1c). Contrary to HNDs, MNDs can be magnetized by external magnetic field. The magnetic moments of MNDs calculated from VSM result is $8.7 \times 10^{-16} \text{ Am}^2$. Therefore, to examine the function of MNDs in neuronal stimulation, the non-magnetic HNDs were used in negative control groups.” (Page 13, line 9 to 19).

Reviewer #2, Comment 3) N numbers in the in vitro and in vivo results should be reported in the paper. For example, in Figure 3 and 4, how many cultures/trials were the reported cell numbers from?

Response: We thank the reviewer for the suggestion. We added the numbers of cells, samples (individual cultures) and animals for each experiment in the legends of main figures and supplementary figures.

Reviewer #2, Comment 4) In the in vivo experiment, the stimulation was made 5-7 days after injection, which is a long time period. It is unclear whether the injected MNDs were still in the STN. The authors should try to show the location and distribution of the MNDs in the tissue (such as fluorescent dye labeling, EM).

Response: We thank the reviewer for this suggestion. To observe the injection location of MNDs, we used alexa-594 to prepare the fluorescence-labeled MNDs for unilateral stereotax injection. At 5 days after injection, we sliced the brain and investigate whether fluorescence-labeled MNDs were still in the brain. We could observe the red fluorescence at the STN after 5 days (Fig. S6c-d). Which indicated that the MNDs were not degraded or metabolized for at least 5 days. The description of this experiment was added to the result section: “*The coordinate of injection was confirmed by using fluorescence-labeled MNDs (Fig. S6a-b). After injection, the fluorescence-labeled MNDs were not degraded or metabolized for at least 5 days (Fig. S6c-d).*” (Page 17, line 3 to line 5)

Reviewer #2, Comment 5) The images of C-fos staining were low quality. The authors should provide higher magnification or better quality images to show the c-Fos and dual staining.

Response: We thank the reviewer for this suggestion. We replaced the original figure with better quality c-fos images. The magnification of c-fos images were also added to Fig 7 and Fig S7.

We thank all reviewers for their constructive comments to further improve the quality of the manuscript on different points.

Yours Sincerely,

Po-Han Chiang, Ph.D.

REVIEWERS' COMMENTS:

Reviewer #1 (Remarks to the Author):

The authors have done a thorough job of revising the manuscript according to the reviewer's comments. All of my review comments have now been addressed. A minor point is that there are a few grammatical and syntax issues that will have to be carefully proofread before the manuscript is published.

Reviewer #2 (Remarks to the Author):

My comments and concerns have been carefully addressed.